# Pat1 promotes processing body assembly by enhancing the phase separation of the DEAD-box ATPase Dhh1 and RNA

Ruchika Sachdev[1], Maria Hondele[1], Miriam Linsenmeier[1], Pascal Vallotton[1], Christopher F Mugler[1,2†], Paolo Arosio[1], Karsten Weis[1]*

[1]ETH Zurich, Zurich, Switzerland; [2]University of California, Berkeley, Berkeley, United States

**Abstract** Processing bodies (PBs) are cytoplasmic mRNP granules that assemble via liquid–liquid phase separation and are implicated in the decay or storage of mRNAs. How PB assembly is regulated in cells remains unclear. Previously, we identified the ATPase activity of the DEAD-box protein Dhh1 as a key regulator of PB dynamics and demonstrated that Not1, an activator of the Dhh1 ATPase and member of the CCR4-NOT deadenylase complex inhibits PB assembly *in vivo* (Mugler et al., 2016). Here, we show that the PB component Pat1 antagonizes Not1 and promotes PB assembly via its direct interaction with Dhh1. Intriguingly, *in vivo* PB dynamics can be recapitulated *in vitro*, since Pat1 enhances the phase separation of Dhh1 and RNA into liquid droplets, whereas Not1 reverses Pat1-Dhh1-RNA condensation. Overall, our results uncover a function of Pat1 in promoting the multimerization of Dhh1 on mRNA, thereby aiding the assembly of large multivalent mRNP granules that are PBs.
DOI: https://doi.org/10.7554/eLife.41415.001

**\*For correspondence:**
karsten.weis@bc.biol.ethz.ch

**Present address:** †Amyris Inc, Emeryville, United States

## Introduction

Cells are often subjected to severe environmental fluctuations such as nutrient deficiency, temperature changes or osmotic shock. To cope with stresses, a variety of mechanisms have evolved allowing cells to respond acutely and survive. These include changes in the gene expression program, which are often driven by robust transcriptional responses. Yet cells must also inactivate old mRNAs that may interfere with adaptation to the new condition, and therefore, post-transcriptional regulation and RNA turnover are critical to enable rapid changes in gene expression (*Ashe et al., 2000*; *Mager and Ferreira, 1993*).

The post-transcriptional fate of eukaryotic mRNAs is tightly linked to the complement of proteins that associate with it to form mRNPs (messenger ribonucleoproteins). For example, the presence of the $m^7$ G cap structure at the 5' end is crucial for eIF4E binding, while Pab1 binds the 3' poly-A tail, both of which protect the mRNA against degradation and promote translation initiation. On the other hand, mRNA turnover is thought to initiate via deadenylation by the Ccr4/Pop2/Not1 complex (*Muhlrad and Parker, 1992*; *Sheth and Parker, 2003*), followed by cap removal by the decapping enzyme Dcp1/2 and 5'−3' degradation of the transcript by the exonuclease Xrn1 (*Coller and Parker, 2004*; *Larimer and Stevens, 1990*). Thus, the processes of mRNA translation and decay are coupled and act antagonistically (*Radhakrishnan et al., 2016*; *Beelman and Parker, 1994*; *Schwartz and Parker, 1999*; *Schwartz and Parker, 2000*; *Sun et al., 2012*; *Hu et al., 2009*; *Chan et al., 2018*)

The balance between mRNA translation and decay is not only temporally but also spatially controlled. For instance in yeast, upon glucose starvation, the majority of cellular translation halts and untranslated mRNA as well as many factors of the RNA decay machinery concentrate in large cytoplasmic mRNP granules, so called processing bodies [PBs] (*Brengues et al., 2005*; *Sheth and*

*Parker, 2003*; *Teixeira et al., 2005*). PBs contain major components of the 5′−3′ RNA decay pathway, for example Dcp1/2 and Xrn1, as well as activators of decapping - namely, the DEAD-box ATPase Dhh1, Pat1, Edc3 and the Lsm1-7 complex. PBs presumably form around mRNAs sequestered away from active translation, and in support of this, inhibition of translation initiation leads to induction of PBs (*Franks and Lykke-Andersen, 2008*; *Chan et al., 2018*; *Teixeira et al., 2005*). On the contrary, inhibition of translation elongation, for example by the drug cycloheximide which traps mRNA on ribosomes, and prevents their influx into PBs, has the opposite effect and leads to PB disassembly (*Teixeira et al., 2005*; *Mugler et al., 2016*).

In consequence, PBs have been implicated in a variety of post-transcriptional processes such as translational repression (*Holmes et al., 2004*; *Coller and Parker, 2005*), mRNA decay (*Sheth and Parker, 2003*; *Cougot et al., 2008*; *Mugler et al., 2016*), mRNA storage (Brengues et al., 2005; *Bhattacharyya et al., 2006*), and in higher eukaryotes, micro RNA-mediated repression (*Liu et al., 2005*; *Pillai, 2005*). PBs and related mRNP granules such as stress granules, P granules in germ cells, neuronal granules etc. form in a number of different species from yeast to mammals, and in a variety of cell types and biological contexts, suggesting that these structures are crucial for cellular function or survival (*Barbee et al., 2006*; *Seydoux and Braun, 2006*). Indeed, there is evidence that PBs can be essential for cell survival under stress. For instance, cells unable to form PBs show a drastic loss in viability in stationary phase (*Ramachandran et al., 2011*; *Shah et al., 2013*). However, the precise role of PBs in the regulation of gene expression is a matter of debate and in order to understand PB function it is critical to examine how PBs assemble and are regulated.

Numerous recent studies have shown that PBs are dynamic, membraneless organelles that assemble from a variety of multivalent but weak RNA-protein, protein–protein and RNA-RNA interactions; a phenomenon called liquid–liquid phase separation (LLPS). The resulting granules are liquid-like, spherical, dynamic and are dissolved by the alcohol 1,6-hexanediol, all key features of LLPS (*Decker et al., 2007*; *Fromm et al., 2012*; *Fromm et al., 2014*; *Guo and Shorter, 2015*; *Mugler et al., 2016*; *Kroschwald et al., 2015*).

Although PB formation occurs via a plethora of multivalent protein–RNA interactions in a redundant fashion (*Rao and Parker, 2017*), there are key players whose deletion drastically attenuates PB assembly (*Decker et al., 2007*; *Pilkington and Parker, 2008*; *Scheller et al., 2007*). One such component is the highly abundant DEAD-box ATPase Dhh1 (DDX6 in humans). Dhh1 is an enhancer of decapping and is involved in the translational repression of mRNAs (*Carroll et al., 2011*; *Coller and Parker, 2005*; *Fischer and Weis, 2002*; *Coller et al., 2001*). We recently showed that Dhh1 undergoes phase separation *in vitro*, and controls PB dynamics *in vivo* via its RNA-stimulated ATPase activity whereas the ATPase activator Not1 dissolves Dhh1 droplets *in vitro* and inhibits PB formation *in vivo*. However, whether additional factors assist Dhh1 in promoting PB formation remains to be determined and is the focus of this study.

One such candidate is the eukaryotic PB component Pat1, an evolutionarily conserved multi-domain RNA binding protein that, like Dhh1, functions in both translational repression and mRNA decay (*Pilkington and Parker, 2008*; *Teixeira and Parker, 2007*). In yeast, Pat1 acts as a scaffold that brings together repression and decay factors (Dhh1, Edc3, Lsm1-7, Dcp1/2 etc.) via direct and/or indirect interactions with mRNPs (*Nissan et al., 2010*). Dhh1 for example directly binds acidic residues in the N-terminal domain of Pat1, an interaction that is conserved across species (*Sharif et al., 2013*; *Ozgur and Stoecklin, 2013*). Deletion of *PAT1* in yeast alters the localization of PB components, whereas its overexpression induces the formation of constitutive PBs even in unstressed cells (*Pilkington and Parker, 2008*; *Coller and Parker, 2005*). Furthermore, it was shown that phosphorylation of Pat1 by the cAMP-dependent protein kinase A (PKA) inhibits PB formation, and cells expressing a phospho-mimetic mutant of Pat1 are defective in PB assembly (*Ramachandran et al., 2011*). However, the mechanism by which Pat1 regulates PB assembly is unclear.

Here, we demonstrate that Pat1 promotes PB formation via Dhh1, and we show that the interaction between Pat1 and Dhh1 is critical for robust PB assembly *in vivo*. Using a liquid phase separation assay, we can recapitulate this function of Pat1 *in vitro* as addition of recombinant Pat1, but not of a Pat1 variant that cannot bind Dhh1, enhances phase separation of Dhh1 and RNA. Overall, our

*in vivo* and *in vitro* data suggests that during stress Pat1 antagonizes the inhibitory effect of Not1 on PB assembly and promotes the multimerization of Dhh1 on mRNA, leading to the formation of PBs.

## Results

### Constitutive PB formation upon Pat1 overexpression is Dhh1 dependent

The PB components Pat1 and Dhh1 were each previously shown to be required for PB formation in yeast (*Pilkington and Parker, 2008*; *Ramachandran et al., 2011*; *Mugler et al., 2016*; *Rao and Parker, 2017*). To understand the relationship between Pat1 and Dhh1 in the regulation of PB assembly, we took advantage of the observation that overexpression of Pat1 leads to constitutive PB formation (*Coller and Parker, 2005*). This allowed us to characterize their respective regulatory contributions without external influences such as nutrient starvation signals. As expected, Pat1 over-expression from its endogenous locus using the galactose promoter in the presence of Dhh1 led to the formation of constitutive PBs, as visualized by co-localization of Dhh1-GFP and Dcp2-mCherry foci (*Figure 1A*). No foci were formed when Pat1 was not overexpressed or when we overexpressed the human influenza hemagglutinin (HA) tag alone (*Figure 1—figure supplement 1A*). Notably, when Pat1 was overexpressed in cells that lack Dhh1, a drastic reduction in the number of PBs was observed, as visualized by the *bona fide* PB marker Edc3-GFP and its co-localization with Dcp2-mCherry (*Figure 1A,B*). This demonstrates that formation of constitutive PBs upon Pat1 overexpression requires Dhh1.

### Pat1 and Dhh1 interaction is essential for PB assembly

Pat1 is an 88 kDa protein that consists of an N-terminal un-structured domain (Pat1-N) followed by a proline-rich stretch plus middle domain and a C-terminal globular fold (Pat1-C). The N-terminal domain of Pat1 directly binds Dhh1's RecA2 core via a conserved DETF motif (*Sharif et al., 2013*; *Ozgur and Stoecklin, 2013*). Furthermore, Pat1's C-terminus harbors two serine residues that are phosphorylated by the cAMP-dependent protein kinase A (PKA) negatively regulating the interaction with Dhh1 and PB formation (*Ramachandran et al., 2011*).

In order to understand how these Pat1 domains influence LLPS and PB assembly, we wanted to characterize various Pat1 mutants both *in vivo* and *in vitro*. Unfortunately, we were unable to recombinantly express functional, full-length Pat1. Based on published crystal structures and *in vitro* interaction studies, we therefore developed a minimal Pat1 construct encompassing both N- and C-terminal domains that we refer to as Pat1-NC [amino acids: 5–79 (N) and 456–787 (C) connected with a (GGS)$_4$ linker] (*Figure 1C*). The N-terminal fragment of Pat1 (amino acids 5–79) was crystallized and shown to directly bind to Dhh1 (*Sharif et al., 2013*) whereas the C-terminus of Pat1 (amino acids 456–787) contains the PKA phosphorylation sites and mediates the interaction with the LSM-complex and RNA (*Sharif et al., 2013*; *Fourati et al., 2014*; *Charenton et al., 2017*; *Ramachandran et al., 2011*; *Pilkington and Parker, 2008*).

To investigate whether this *PAT1-NC* construct is functional *in vivo* and behaves similar to wild-type Pat1, we took advantage of the temperature-sensitive growth defect of a *PAT1* deletion mutant (*Fourati et al., 2014*). Like full-length Pat1, expression of the *PAT1-NC* construct fully restored growth of *pat1Δ* cells at 37°C (*Figure 1—figure supplement 1B and C*). In addition, we also tested if the expression of *PAT1-NC* is able to rescue the synthetic lethality that was observed in *pat1Δ edc3Δ scd6Δ* cells (*Fourati et al., 2014*). The expression of *PAT1-NC* rescued the growth phenotype of *pat1Δ edc3Δ scd6Δ* cells at 37°C to a similar extent as achieved by wild-type *PAT1* (*Figure 1—figure supplement 1D*).

Furthermore, cells expressing the *PAT1-NC* construct when starved of glucose induced PBs to a similar level as wild-type *PAT1* (*Figure 1D,E*) indicating full functionality *in vivo*. The PBs in the *PAT1-NC* expressing cells rapidly dissolved upon re-addition of glucose demonstrating that they are reversible and dynamic rather than anomalous granules (*Figure 1D*). In addition, Pat1-NC-GFP itself localized to PBs upon stress in a reversible manner (*Figure 1—figure supplement 1E*). Finally, when we overexpressed Pat1-NC using a GAL promoter, robust PBs were induced in the absence of stress (*Figure 1F,G*). Taken together our data suggest that the *PAT1-NC* construct is fully functional *in vivo*.

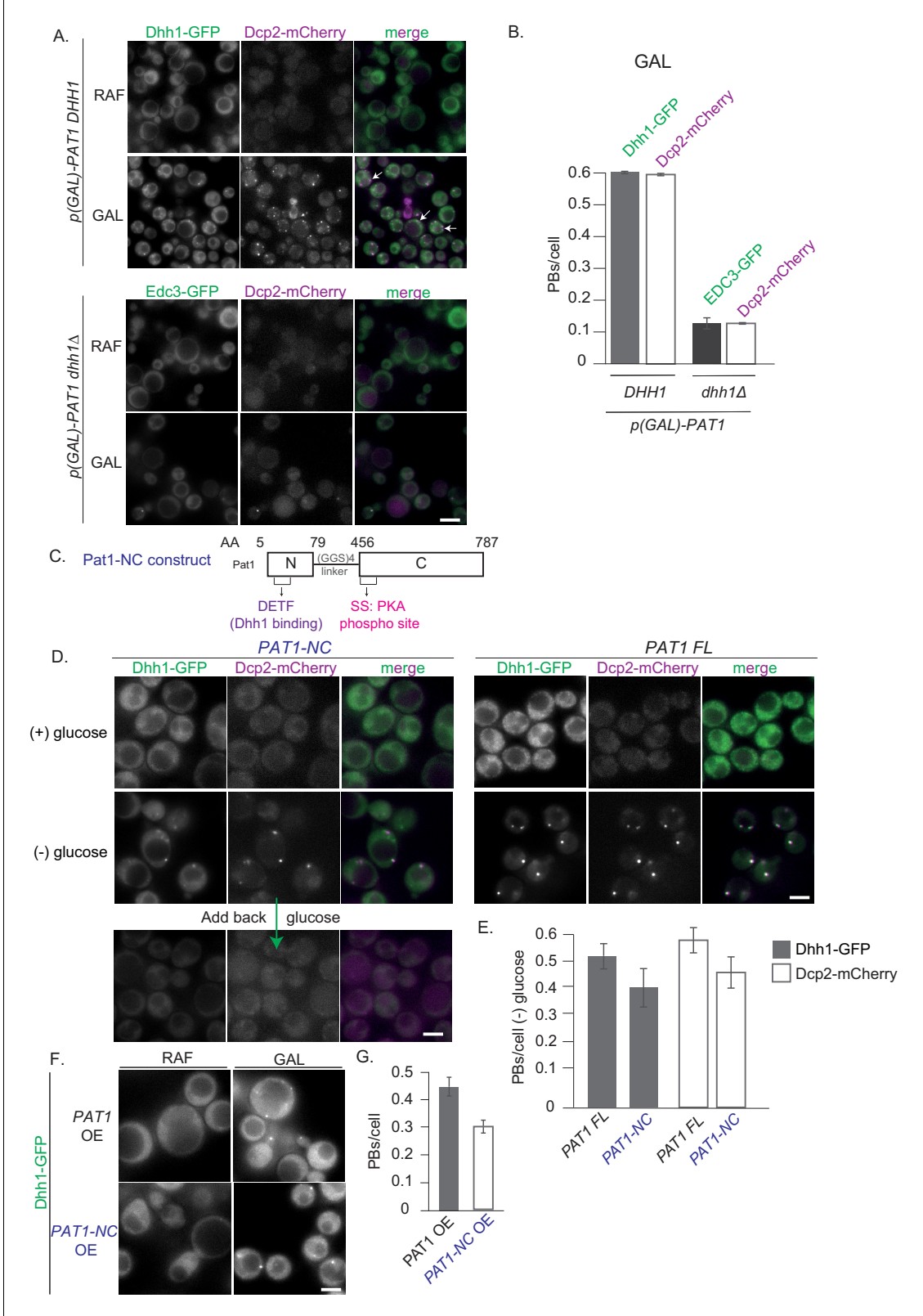

**Figure 1.** Constitutive PB formation upon Pat1 overexpression is Dhh1 dependent. (**A**) Overexpression (OE) of Pat1 leads to constitutive PB formation but only in the presence of Dhh1. Cells co-expressing the indicated PB components were grown in synthetic complete (SC) raffinose media to exponential growth phase after which Pat1 was overexpressed via addition of galactose. Cells in both raffinose and galactose were observed by fluorescence microscopy. In all Pat1 OE strains the endogenous promoter of Pat1 was replaced by the galactose 1–10 promoter (*p-GAL-PAT1*). Scale

*Figure 1 continued on next page*

Figure 1 continued

bar: 5 µm. (B) Quantification of images in A depicting number of PBs/cell. N = 3 biological replicates with >800 cells/replicate. Error bars: SEM. (C) Pat1-NC (AA 5–79 + 456–587) is functional *in vivo*. Cartoon of the Pat1-NC construct (see text for details). (D) *PAT1-NC* induces PB formation upon stress *in vivo*. Cells co-expressing the indicated the PB components in *PAT1* full length or *PAT1-NC* background were grown in synthetic complete (SCD) media to exponential growth phase then shifted to glucose-rich or glucose-starvation conditions for 30 min and observed by fluorescence microscopy. PBs induced in the *PAT1-NC* background were dissolved by addition of 2% glucose demonstrating reversibility. Scale bar: 5 µm. (E) Quantification of images shown in A depicting number of PBs/cell. Bars: SEM. N = 3 biological replicates with >800 cells/replicate. (F) *PAT1-NC* (AA 5–79 + 456–587) OE leads to constitutive PB induction. *p(GAL)-PAT1* or *p(GAL)-PAT1-NC* cells expressing Dhh1-GFP were grown in SC raffinose media to exponential growth phase after which Pat1 OE was induced with galactose. Scale bar: 5 µm. (G) Graph depicts Dhh1 PBs/cell, SEM. N = 3 biological replicates with >800 cells/replicate. [Diatrack 3.05 and cell segmentation using in house Matlab code was used for the quantification of PB and cell numbers respectively for all the experiments shown in the manuscript].

DOI: https://doi.org/10.7554/eLife.41415.002

The following figure supplement is available for figure 1:

**Figure supplement 1.** Controls for Pat1 OE related to Figure 1 and growth rescue experiments of Pat1-NC.

DOI: https://doi.org/10.7554/eLife.41415.003

We next examined whether Pat1-NC can interact with Dhh1 *in vitro*, and as expected, Pat1-NC displayed strong binding to full-length Dhh1 in a GST pull down assay (*Figure 2A*). However, this interaction was completely abolished for a Pat1-NC variant in which the N-terminal DETF motif was mutated to four alanines [Pat1^4A-Dhh1, see *Supplementary file 2* table S2B for a list of all mutants used in this study] (*Figure 2A*).

We next investigated the effect of the *pat1^4A-Dhh1* mutant in the context of the full-length protein on PB formation *in vivo*. While cells expressing *pat1^4A-Dhh1* showed a mild growth defect compared to the wild type *PAT1*, they grew significantly better than *pat1Δ* cells, indicating that this protein is at least partially functional (*Figure 2—figure supplement 1A*). To examine whether the interaction between Pat1 and Dhh1 is required for PB formation, we monitored PB assembly upon stress in the absence of glucose in wild-type (*PAT1*) or the *pat1^4A-Dhh1* background. As expected, the number of PBs was drastically reduced in the complete absence of Pat1, as visualized by Dhh1-GFP (*Figure 2—figure supplement 1B*). Importantly, *pat1^4A-Dhh1* expressing cells also demonstrated a drastic reduction in PB number compared to cells expressing *PAT1* (*Figure 2B,C*) and the Pat1^4A-Dhh1-GFP protein itself showed a significant defect in localizing to PBs (*Figure 2D,E*).

To assure that the Pat1^4A-Dhh1 mutant is stably expressed, we checked its protein levels using Western blotting and observed that the mutant expresses to similar levels as wild-type Pat1 (*Figure 2—figure supplement 1C*). Moreover, in the Pat1^4A-Dhh1 mutant cells, the expression levels of Dhh1 were comparable to the Dhh1 levels in cells harboring a wild-type copy of Pat1 (*Figure 2—figure supplement 1D*). This demonstrates that the defect in PB formation in cells expressing the Pat1^4A-Dhh1 binding mutant is specifically due to impaired Pat1 function.

We were unable to overexpress the *Pat1^4A-Dhh1* mutant in yeast. In order to investigate PB formation independently of carbon starvation stress, we therefore treated cells with the drug hippuristanol, which inhibits the eukaryotic translation initiation factor eIF4A (Bordeleau, M.E., et al., 2006). In consequence, hippuristanol prevents translation initiation and robustly induces PB formation within minutes in cells expressing wild-type *PAT1* (*Chan et al., 2018*) [*Figure 2F*, *Video 1*]. Cells expressing *pat1^4A-Dhh1* however, did not show PB formation until 2 hrs after hippuristanol addition (*Figure 2F* and *Video 2*).

It cannot be excluded that the Pat1^4A-Dhh1 mutant might interfere with the binding of Pat1 to some other factor with a role in PB formation in addition to Dhh1. We therefore also employed a complementary approach wherein we examined mutants on Dhh1 that impair interaction with Pat1. Two such Dhh1 mutants have been described: Dhh1 Mut3A (Dhh1^S292DN294D) and Dhh1 Mut3B (Dhh1^R295D) that partially or completely abolish binding to Pat1, respectively (*Sharif et al., 2013*). Cells expressing either wild-type Dhh1, Dhh1^S292DN294D, Dhh1^R295D or lacking Dhh1 altogether were starved of glucose and PB assembly was assessed. Whereas the Dhh1 variants expressed as well as wild-type Dhh1 (*Figure 3—figure supplement 1*), we observed a drastic reduction in the number of PBs in both Dhh1^S292DN294D, Dhh1^R295D cells (*Figure 3A,B*), and as published before also in *dhh1Δ* cells (*Figure 3A,B*, Mugler et al., 2016). The Dhh1^R295D had a more severe PB formation defect

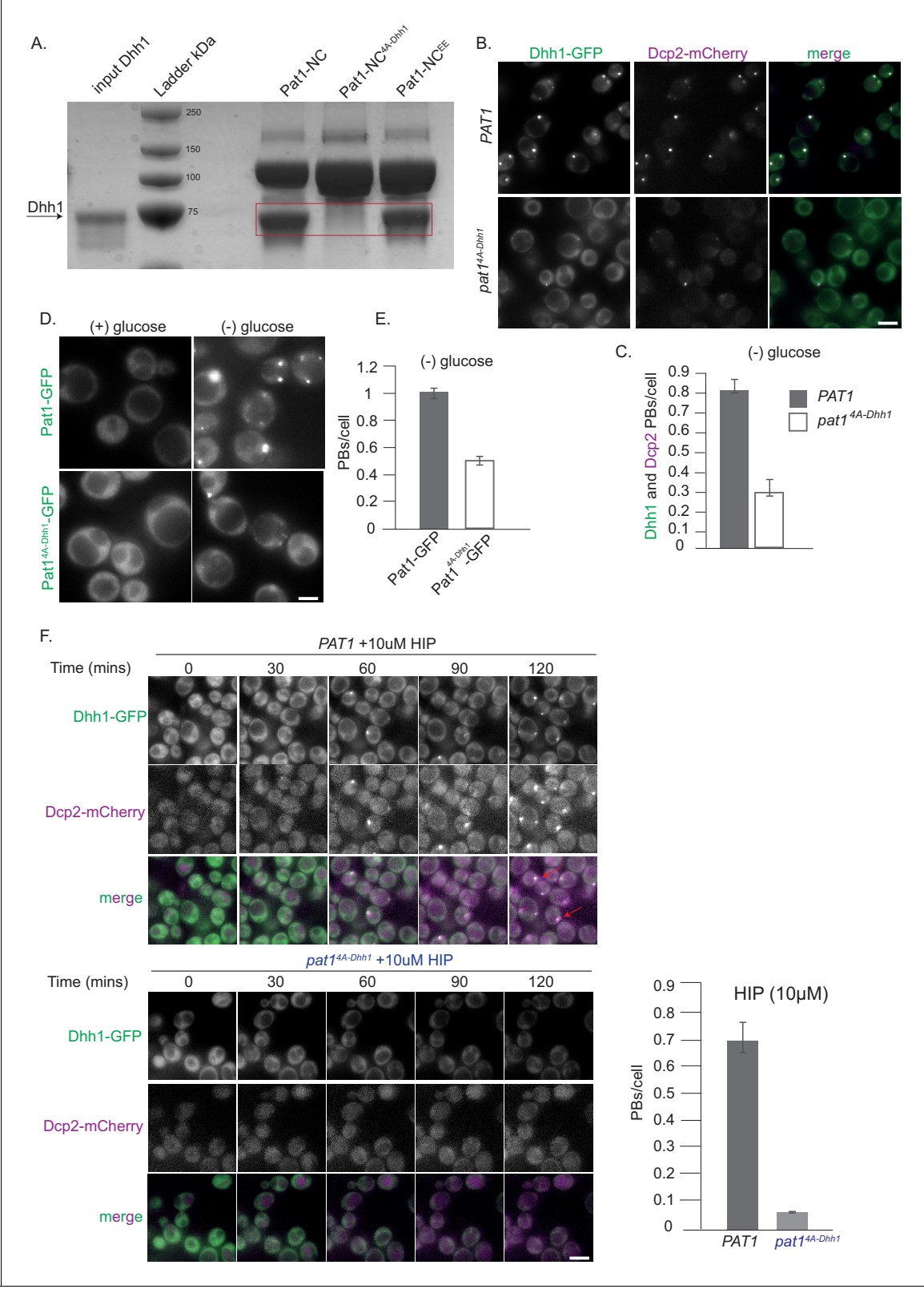

**Figure 2.** Pat1-Dhh1 interaction is essential for PB assembly. (**A**) Recombinant Pat1-NC[4A-Dhh1] is defective for Dhh1 binding. GST pull-down with GST-Pat1-NC and mutants thereof as matrix-bound bait and Dhh1 as prey. After 1 hr of binding at 4°C, the samples were washed five times with a buffer containing 300 mM NaCl and visualized by Coomassie staining after separation on a 12% acrylamide gel. (**B**) Expression of *pat1* [4A-Dhh1] (Dhh1 binding mutant) leads to a drastic reduction in PB formation. Cells co-expressing the indicated PB components in the *PAT1* or *pat1[4A-Dhh1]* background were

*Figure 2 continued on next page*

Figure 2 continued

grown in SCD media to exponential growth phase, then shifted to glucose-rich or glucose-starvation conditions for 30 min and observed by fluorescence microscopy. Scale bar: 5 μm. (C) Quantification of images in B depicting number of PBs/cell, (Dhh1-GFP PBs co-localizing with Dcp2-mCherry) are shown, three biological replicates, bars: SEM. (D) Pat1$^{4A-Dhh1}$-GFP mutant is defective for PB localization. Cells expressing either Pat1-GFP or the Pat1$^{4A-Dhh1}$-GFP mutant were grown in SCD media to exponential growth phase then shifted to glucose-rich or glucose-starvation conditions for 30 min and observed by fluorescence microscopy. Scale bar: 5 μm. (E) Quantification of images shown in D depicting number of PBs/cell. N = 3 biological replicates with >500 cells/replicate. Error bars: SEM. (F) pat1$^{4A-Dhh1}$ mutant is defective in PB formation upon hippuristanol treatment. Cells expressing either PAT1 or the pat1$^{4A-Dhh1}$ were grown in SCD media to exponential growth phase after which they were treated with 10 μM hippuristanol for 2 hrs. The kinetics of PB formation was monitored using Dhh1-GFP and its co-localization with Dcp2-mCherry. Quantification of PBs/cell is depicted. Stills from the live imaging time course are shown. N = 2 biological replicates, two technical replicates each with >300 cells/replicate, SD.

DOI: https://doi.org/10.7554/eLife.41415.004

The following figure supplement is available for figure 2:

**Figure supplement 1.** Expression levels of various Pat1 mutants used in this study and their growth phenotypes.

DOI: https://doi.org/10.7554/eLife.41415.005

compared to the Dhh1$^{S292DN294D}$ aligning with the *in vitro* binding abilities of these Dhh1 mutants to Pat1 (*Sharif et al., 2013*).

To test PB formation independently of carbon starvation stress in these Pat1-binding mutants of Dhh1, we used the drug hippuristanol as before. Dhh1 wild-type cells when treated with hippuristanol formed PBs within minutes as judged by the co-localization of Dhh1-GFP and Dcp2-mCherry foci (*Figure 3C,D* and *Video 3*). However, both the Dhh1$^{S292DN294D}$ and Dhh1$^{R295D}$ mutants did not form robust PBs until 3 hrs of hippuristanol treatment exhibiting a phenotype similar to *dhh1Δ* cells (*Figure 3C,D* and *Videos 4*, *5* and *6*). Taken together, our data reveal that the Pat1-Dhh1 interaction is pivotal for robust PB assembly.

## Pat1 phosphorylation status influences RNA binding

Pat1 is a target of the cAMP-dependent protein kinase A (PKA) and phosphorylation of two serines (amino acids 456/457) in the C-terminus of Pat1 under glucose-rich conditions was previously shown to negatively regulate PB formation (*Ramachandran et al., 2011*). In agreement with the published literature, cells expressing the *pat1$^{EE}$* mutant showed a defect in PB formation (*Figure 4—figure supplement 1A,B* and *Ramachandran et al., 2011*). To better understand how Pat1 phosphorylation controls PB assembly, we sought to characterize the influence of the phosphorylation status of Pat1 on PB formation in the absence of stress. Overexpression of *PAT1$^{WT/SS}$* (WT = wild type) and *pat1$^{AA}$* (non-phosphorylatable) led to constitutive PB formation as visualized by Dhh1-GFP and Dcp2-mCherry positive foci. However, upon overexpression of *pat1$^{EE}$* (phospho-mimetic), foci number was reduced (*Figure 4A,B*). The Pat1$^{EE}$-GFP mutant was expressed as well as wild-type Pat1-GFP and had no effect on Dhh1 expression (*Figure 2—figure supplement 1C and D*) emphasizing that the phospho-mimetic mutant specifically impairs Pat1 function.

Pull-down experiments from cell extracts previously suggested that PKA-dependent phosphorylation of Pat1 diminishes its interaction with Dhh1 (*Ramachandran et al., 2011*). To test whether this effect is direct, we performed pull-down assays with the recombinant Pat1-NC protein and the Pat1-NC$^{EE}$ variant. However, in this assay there was no significant difference in Dhh1 binding between Pat1-NC and Pat1-NC$^{EE}$ (*Figure 2A*). Since the strong DETF binding site in Pat1-N might mask any weaker interaction in Pat1-C, we also tested for direct binding of Pat1-C to Dhh1 but were unable to detect any interaction (*Figure 4C*), indicating that the DETF-motif in Pat1-N provides the major interaction surface for Dhh1.

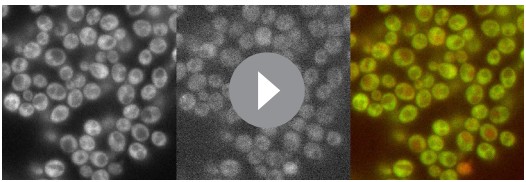

**Video 1.** Hippuristanol induces PBs in cells expressing wild-type *PAT1*. The video depicts PB formation (Dhh1-GFP and Dcp2-mCherry co-localization) in cells expressing wild-type *PAT1* upon hippuristanol treatment. (2 hr video, 5 min intervals; video played at seven fps). Each frame is a single plane.

DOI: https://doi.org/10.7554/eLife.41415.006

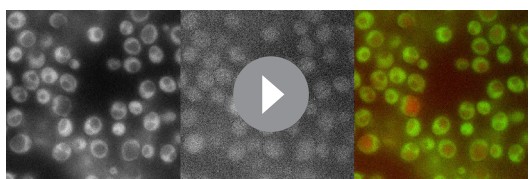

**Video 2.** Hippuristanol fails to induce PBs in cells expressing *pat1⁴ᴬ⁻ᴰʰʰ¹* mutant. The video depicts failure to form PBs (Dhh1-GFP and Dcp2-mCherry co-localization) in cells expressing *pat1⁴ᴬ⁻ᴰʰʰ¹* upon hippuristanol treatment. (2 hr video, 5 min intervals; video played at seven fps). Each frame is a single plane.
DOI: https://doi.org/10.7554/eLife.41415.007

Pat1-C was previously shown to bind RNA (*Pilkington and Parker, 2008*; *Chowdhury et al., 2014*). We therefore tested whether RNA binding is regulated by PKA-dependent phoshphorylation by performing RNA oligo gel shift assays using recombinant proteins. Whereas wild-type Pat1-C robustly binds the RNA oligo and shifts it to higher molecular weight in a native PAGE gel, Pat1ᴱᴱ displayed a significantly reduced RNA shift (quantified in *Figure 4D*). Taken together, our *in vivo* and *in vitro* results on the phospho-mimetic Pat1ᴱᴱ mutant suggest that in addition to Pat1-Dhh1 binding, also the interaction between Pat1 and RNA is important for PB formation.

## Pat1 and Not1 act antagonistically *in vivo*

Our lab previously showed that Not1 promotes PB disassembly by activating the ATPase activity of Dhh1, and cells expressing a Not1 mutant that cannot bind to Dhh1 (*not1⁹ˣ⁻ᴰʰʰ¹*) form constitutive PBs in non-stressed cells. Yet these PBs are less intense and fewer in number than those formed upon glucose starvation (*Mugler et al., 2016*), indicating that additional mechanisms prevent PB formation in glucose-replete conditions.

Since in glucose-rich media PB assembly is also inhibited by the PKA-dependent phosphorylation of Pat1 (*Ramachandran et al., 2011*), we wanted to test whether expression of the non-phosphorylatable *pat1ᴬᴬ* variant enhances PB formation in the *not1⁹ˣ⁻ᴰʰʰ¹* strain background. While expression of *pat1ᴬᴬ* alone was not sufficient to induce constitutive PBs as observed before (*Ramachandran et al., 2011*) and *Figure 4—figure supplement 1A,B*), expression of *pat1ᴬᴬ* in the presence of *not1⁹ˣ⁻ᴰʰʰ¹* significantly increased the number of constitutive PBs compared to cells expressing *not1⁹ˣ⁻ᴰʰʰ¹* alone (*Figure 5A and B*). Yet, these PBs are still not as bright and numerous as in stressed wild-type cells. Interestingly, PB intensity strongly increased when these strains were further treated with hippuristanol, a drug that blocks initiation and liberates mRNA molecules from polysomes. The percentage of large PBs was highest in the *pat1ᴬᴬ + not19⁹ˣ⁻ᴰʰʰ¹* strain, followed by the *not1⁹ˣ⁻ᴰʰʰ¹* and then *pat1ᴬᴬ* (*Figure 5A,B*).

Overall, this suggests that there are at least three determinants that act cooperatively to enhance the formation of PBs: (a) direct interactions between Pat1-Dhh1 and Pat1-RNA; (b) lack of Not1 binding to Dhh1 and in consequence low Dhh1 ATPase activity (*Mugler et al., 2016*); and (c) the availability of PB client mRNAs that are not engaged in translation. Furthermore, our results suggest that Pat1 and Not1 act antagonistically on PB dynamics. Whereas Pat1 binding to Dhh1 enhances the assembly of PBs, the interaction of Not1 with Dhh1 promotes PB disassembly, likely through activation of the ATPase activity of Dhh1.

## Pat1 promotes PB assembly by enhancing the phase separation of Dhh1 and RNA

At least two mechanisms of how Pat1 promotes PB formation via Dhh1 could be envisioned: first, Pat1 could slow the ATPase cycle of Dhh1, for example by preventing Not1 binding and/or inhibiting ATP hydrolysis, second, Pat1 could directly promote oligomerization and condensation of Dhh1 on RNA.

To test the first hypothesis, we used an *in vitro* ATPase assay using purified protein components in which Not1 robustly stimulates the ATPase activity of Dhh1 (*Mugler et al., 2016*). However, we did not observe a significant inhibition of the Not1-stimulated ATPase activity upon addition of Pat1-NC (*Figure 6—figure supplement 1A*), suggesting that Pat1 does not directly interfere with the mechanism of ATPase activation.

We had previously shown that PB dynamics are regulated by the ATPase activity of Dhh1 (*Carroll et al., 2011*; *Mugler et al., 2016*). To test Pat1's effect on the Dhh1 ATPase cycle *in vivo*,

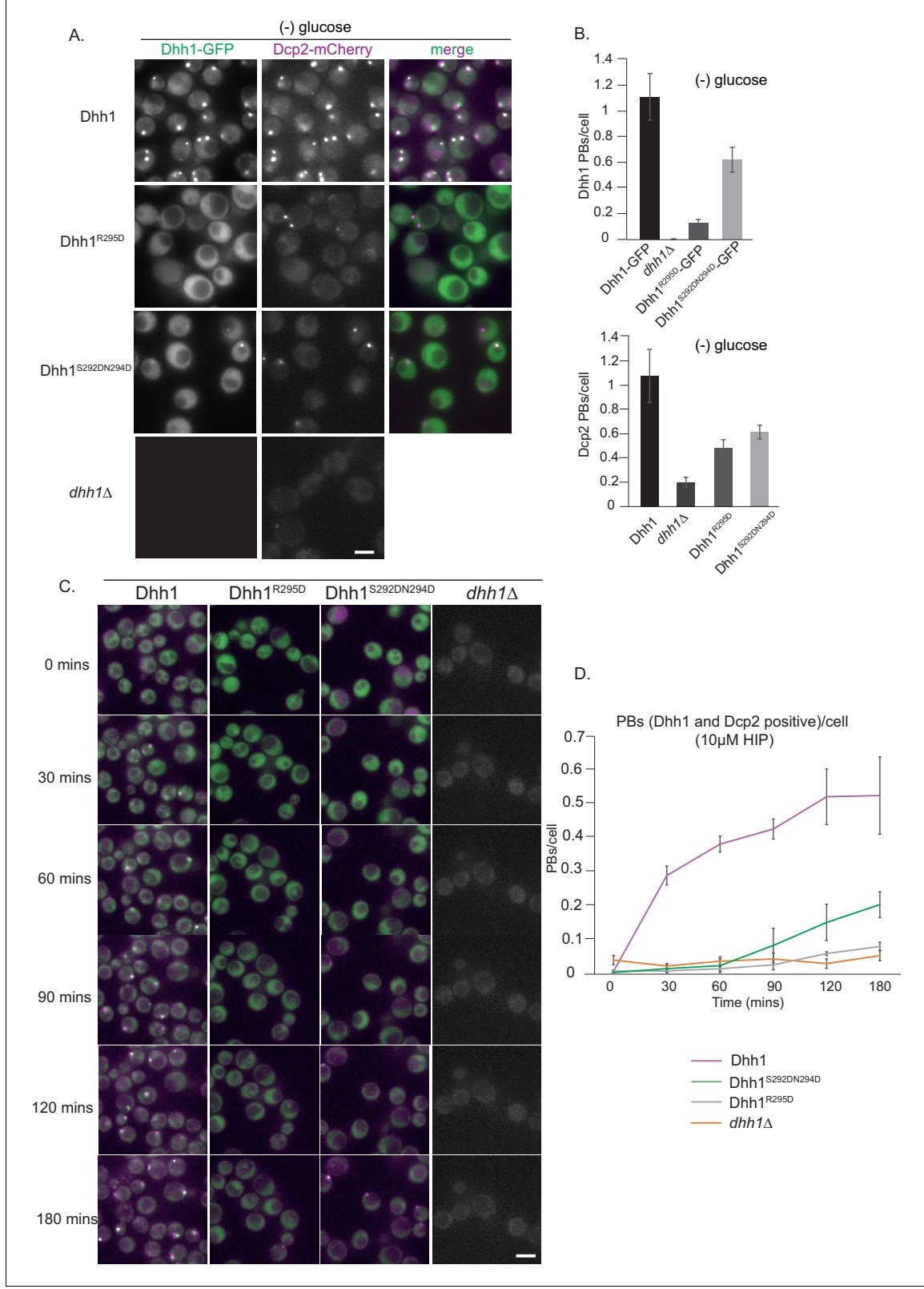

**Figure 3.** Dhh1$^{R295D}$ and Dhh1$^{S292DN294D}$ (Pat1 binding mutants) are defective in PB formation. (**A**) Expression of two distinct Pat1 binding mutants, Dhh1$^{R295D}$ and Dhh1$^{S292DN294D}$, leads to a drastic reduction in PB formation. Cells co-expressing the indicated PB components in either Dhh1, Dhh1$^{R295D}$ or Dhh1$^{S292DN294D}$ cells were grown in SCD media to exponential growth phase, then shifted to glucose-rich or glucose-starvation conditions for 30 min and observed by fluorescence microscopy. Scale bar: 5 µm. (**B**) Quantification of images in A depicting number of Dhh1 positive and Dcp2

*Figure 3 continued on next page*

*Figure 3 continued*

positive PBs/cell; N = 4 biological replicates with >300 cells/replicate, SEM. (**C**) Dhh1$^{R295D}$ and Dhh1$^{S292DN294D}$ mutants are defective in PB formation upon hippuristanol treatment. Cells expressing either *Dhh1,* Dhh1$^{R295D}$, Dhh1$^{S292DN294D}$ or *dhh1Δ*, were grown in SCD media to exponential growth phase after which they were treated with 10 µM hippuristanol for 3 hrs. The kinetics of PB formation was monitored using Dhh1-GFP and its co-localization with Dcp2-mCherry. (**D**) Quantification of PBs/cell. Stills from the live imaging time course are shown. N = 3 biological replicates with >300 cells/replicate, SEM.

DOI: https://doi.org/10.7554/eLife.41415.008

The following figure supplement is available for figure 3:

**Figure supplement 1.** Dhh1 mutants that abolish Pat1 binding are expressed at similar levels to wild-type Dhh1.

DOI: https://doi.org/10.7554/eLife.41415.009

we therefore also examined the turnover of PBs that form upon Pat1 overexpression. The drug cycloheximide (CHX) traps mRNAs on polysomes and thereby stops the supply of new RNA clients to PBs. Since active Dhh1 constantly releases mRNA molecules from PBs, a lack of mRNA influx eventually leads to their disassembly and in consequence the catalytic dead mutant of Dhh1 that is locked in the ATP- state (Dhh1$^{DQAD}$) inhibits PB disassembly and was shown to have negligible turn-over rates (*Mugler et al., 2016*; *Kroschwald et al., 2015*). We observed that PBs formed upon Pat1 overexpression were more dynamic than PBs formed in the presence of Dhh1$^{DQAD}$ and their disassembly kinetics was comparable to PBs formed upon stress in a wild-type *DHH1* background (*Figure 6—figure supplement 1B,C* and *Videos 7*, *8* and *9*). Thus, overall, our *in vitro* and *in vivo* findings are not consistent with the conclusion that Pat1 blocks the ATPase activation of Dhh1.

The second hypothesis is that Pat1 promotes higher-order mRNP assembly by directly or indirectly promoting Dhh1 oligomerization, thereby acting as a scaffold and providing additional protein–protein or protein-RNA interactions (*Coller and Parker, 2005*; *Rao and Parker, 2017*). We have previously shown that recombinant Dhh1 can undergo LLPS in the presence of RNA and ATP, and that these Dhh1 droplets can recapitulate aspects of *in vivo* PB dynamics. (*Mugler et al., 2016*). We therefore utilized this *in vitro* system to test Pat1's impact on the phase separation behavior of Dhh1.

Dhh1 droplets were assembled from purified components as described previously (*Mugler et al., 2016*). Interestingly, while no LLPS of Pat1-NC alone was detected (*Figure 6—figure supplement 2A*), addition of increasing concentrations of wild-type Pat1-NC strongly enhanced the LLPS of Dhh1 in the presence of ATP and RNA as judged by an increase in the area * intensity of Dhh1 droplets (*Figure 6A*). Furthermore, Pat1-NC itself enriched in Dhh1 droplets, suggesting that Pat1 and Dhh1 co-oligomerize with RNA to form a composite phase-separated compartment (*Figure 6A*).

In order to test if the enhancement in phase separation of Dhh1 via Pat1 is RNA-dependent, we performed this experiment also in the absence of RNA. Interestingly, we did not observe any Dhh1 droplets upon addition of increasing concentrations of Pat1-NC when RNA was omitted from the reaction (*Figure 6—figure supplement 2B*) suggesting that under these *in vitro* conditions RNA plays a critical role in the Pat1-promoted enhancement of Dhh1 oligomerization and higher-order phase separation. Consistent with this, the Pat1-N- terminus alone which cannot bind to RNA also does not enhance the phase separation of Dhh1 even in the presence of RNA and ATP (*Figure 6—figure supplement 2C*).

Overall, our *in vitro* data suggests that a combination of both the Pat1-N (binding to Dhh1) and Pat1-C (binding to RNA) is critically required to enhance the phase separation of Dhh1 and promote higher-order oligomeric mRNP structures akin to *in vivo* PBs.

To test the specificity of LLPS stimulation, we tested the Pat1$^{4A-Dhh1}$ (Dhh1 binding) and Pat1$^{EE}$ (phospho-mimetic) mutants both of which drastically reduce PB formation *in vivo* (see *Figures 2B* and *4A,B* and *Figure 4—figure supplement 1A*). Consistent with the *in vivo* results,

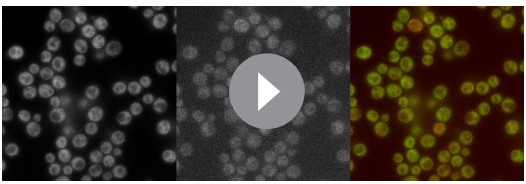

**Video 3.** Hippuristanol induces PBs in cells expressing wild-type *Dhh1*. The video depicts PB formation (Dhh1-GFP and Dcp2-mCherry co-localization) in cells expressing wild-type Dhh1 upon hippuristanol treatment. (3 hr video, 5 min intervals; video played at seven fps). Each frame is a single plane.

DOI: https://doi.org/10.7554/eLife.41415.010

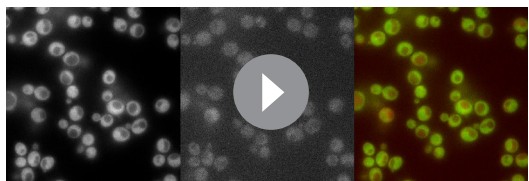
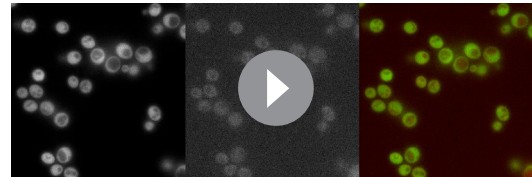

**Video 4.** Hippuristanol does not induce PBs in cells expressing the Dhh1$^{S292DN294D}$ mutant. The video depicts PB formation (Dhh1$^{S292DN294D}$-GFP and Dcp2-mCherry co-localization) in cells expressing the Dhh1$^{S292DN294D}$ mutant upon hippuristanol treatment. (3 hr video, 5 min intervals; video played at seven fps). Each frame is a single plane.
DOI: https://doi.org/10.7554/eLife.41415.011

**Video 5.** Hippuristanol fails to induce PBs in cells expressing the Dhh1$^{R295D}$ mutant. The video depicts PB formation (Dhh1$^{R295D}$-GFP and Dcp2-mCherry co-localization) in cells expressing the Dhh1$^{R295D}$ mutant upon hippuristanol treatment. (3 hr video, 5 min intervals; video played at seven fps). Each frame is a single plane.
DOI: https://doi.org/10.7554/eLife.41415.012

Dhh1-RNA droplet formation was either not at all or only mildly stimulated upon addition of these Pat1 variants (*Figure 6A*).

We previously demonstrated that the MIF4G domain of Not1 prevents formation of Dhh1-RNA droplets, presumably by activating the ATPase cycle of Dhh1 (*Mugler et al., 2016*). We therefore next analyzed whether the Pat1-Dhh1 droplets are still responsive to Not1. Similar to what we observed *in vivo* (*Figure 5*), Not1$^{MIF4G}$ also diminishes formation Dhh1 droplets in the presence of Pat1 *in vitro* (*Figure 6B*).

Taken together our results suggest that Pat1 enhances the multivalency of protein–protein and protein-RNA interactions in a Dhh1-Pat1 mRNP and in consequence promotes the formation of higher-order, liquid-like mRNP droplets akin to *in vivo* PBs.

## *In vitro* phase separated droplets mimic the stoichiometry of PB components *in vivo*

To examine how well our *in vitro* droplets resemble *in vivo* PBs we investigated the stoichiometric ratio of Dhh1:Pat1 in these granules. In order to determine the stoichiometry *in vivo*, we measured the number of PBs and the fluorescence intensity of GFP-tagged Dhh1 and Pat1 in these foci. Briefly, we glucose starved both Pat1-GFP and Dhh1-GFP expressing cells to induce PBs (*Figure 7—figure supplement 1A*) and used the single particle tracking software Diatrack to count the number and intensity of PBs in an unbiased and automated manner in each strain (*Vallotton et al., 2017*). Surprisingly, despite the fact that the cellular concentration of Pat1 is ten-times lower than Dhh1 (*Figure 7—figure supplement 1B*), the ratio of the two PB components was approximately 2:1 ± 0.18 (Dhh1:Pat1) after 0.5–4 hr of starvation (*Figure 7A*). Extended starvation resulted in enhanced Dhh1 recruitment to PBs until reaching a Dhh1:Pat1 ratio of 2.5:1 ± 0.23, suggesting that PB composition matures over time (*Figure 7A*).

In order to determine the stoichiometry of Pat1 and Dhh1 droplets *in vitro*, we imaged Dhh1-mCherry and Pat1-GFP droplets with a confocal microscope (*Figure 7—figure supplement 1C*). The protein concentration of both Dhh1 and Pat1 in the droplet was calculated

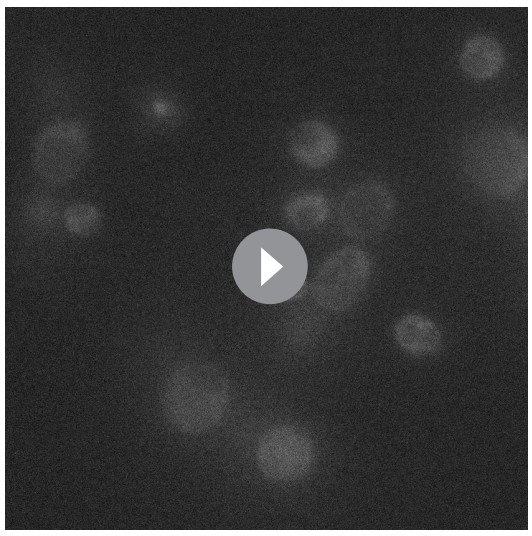

**Video 6.** *dhh1Δ* cells do not form PBs upon hippuristanol treatment. The video depicts PB formation (Dcp2-mCherry co-localization) in cells lacking *dhh1* upon hippuristanol treatment. (3 hr video, 5 min intervals; video played at seven fps). Each frame is a single plane.
DOI: https://doi.org/10.7554/eLife.41415.013

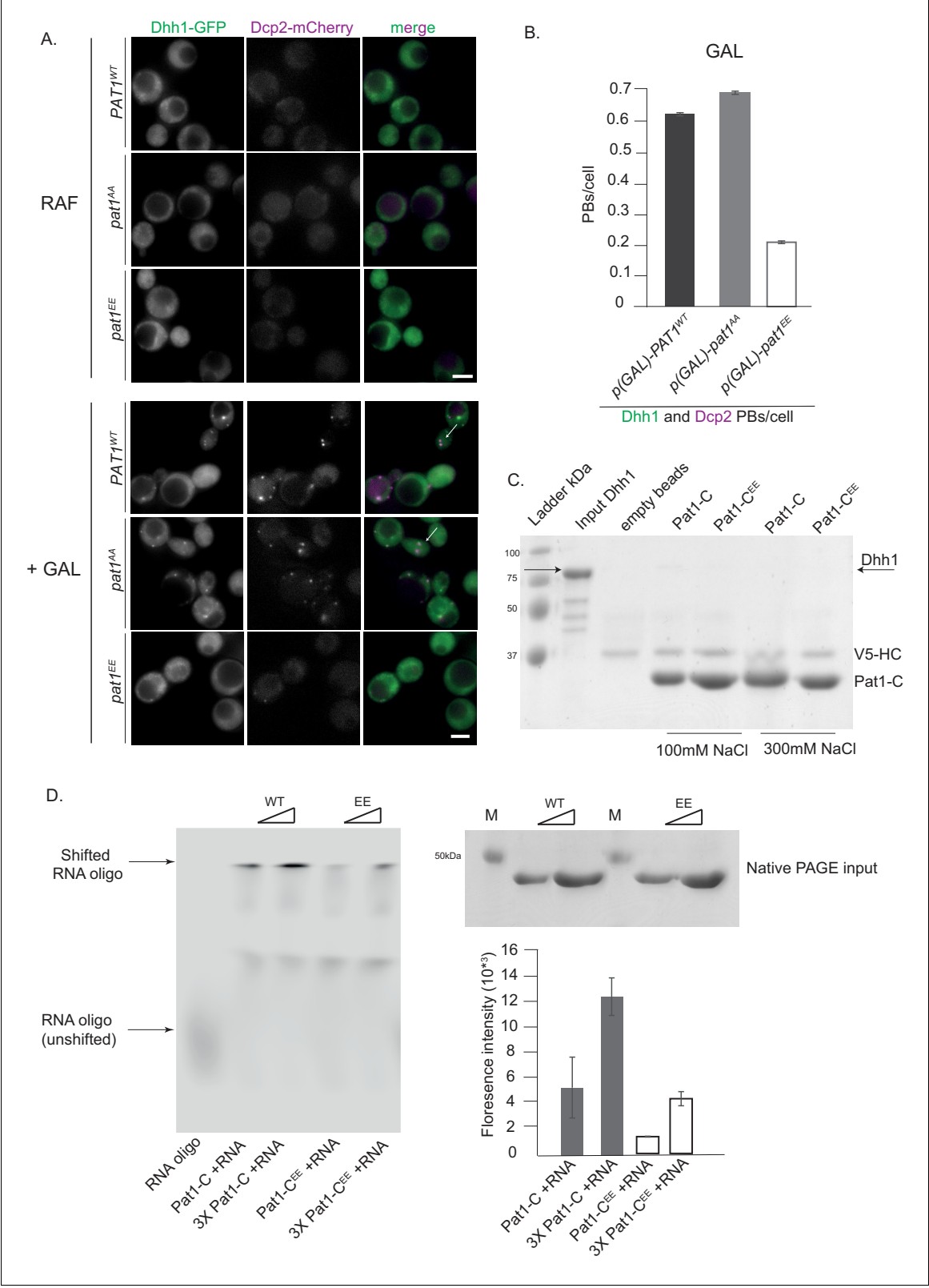

**Figure 4.** Overexpression (OE) of *PAT1^WT* (wild-type) and *pat1^AA* (non- phosphorylatable) leads to constitutive PB formation but OE of *pat1^EE* (phospho-mimetic) does not and the Pat1-C^EE mutant weakens Pat1-RNA binding. (**A**) Cells co-expressing the indicated PB components were grown in SC raffinose media to exponential growth phase after which OE of different Pat1 alleles was induced by galactose addition. Cells were imaged using fluorescence microscopy. (**B**) Quantification of images shown in A depicting number of PBs/cell. Scale bar: 5 μm, SEM. N = 3 biological

*Figure 4 continued on next page*

*Figure 4 continued*

replicates with >800 cells/replicate. (**C**) Pat1-C-terminal domain does not bind Dhh1. V5 pull-down with V5-Pat1-C and V5-Pat1-C$^{EE}$ as matrix-bound bait and Dhh1-mCherry as prey. After 1 hr of binding at 4°C, the samples were washed four times with a buffer containing NaCl as indicated and visualized by Coomassie staining after separation on a 12% acrylamide gel. (**D**) Pat1-C$^{EE}$ weakens the interaction with RNA compared to Pat1-C. Recombinant Pat1-C (WT) and Pat1-C$^{EE}$ were incubated with a Cy5-labeld 29nt RNA oligo in the presence of 300 mM NaCl and separated on a NativePage gradient gel. The fluorescence of the oligo was detected by LI-COR imaging. Reactions were performed in triplicate and the intensity of the upper shifted RNA oligo band was quantified. Equal input of the Pat1-C proteins was verified by acrylamide PAGE and Coomassie staining. Error bars: SD.

DOI: https://doi.org/10.7554/eLife.41415.014

The following figure supplement is available for figure 4:

**Figure supplement 1.** *PAT1* and *pat1$^{AA}$* lead to PB induction upon stress, which is drastically reduced in the *pat1$^{EE}$* background.
DOI: https://doi.org/10.7554/eLife.41415.015

from a standard curve determined from values measured for different concentrations of the respective soluble fluorophores (*Figure 7B*). Remarkably, we found a stoichiometric ratio of 2.7:1 ± 0.13 of Dhh1 to Pat1 in the *in vitro* phase-separated droplets, closely resembling the Dhh1:Pat1 ratio of mature PBs *in vivo* (*Figure 7B*).

Thus, while other mRNA decay factors also contribute to PB formation *in vivo*, our data demonstrate that with a minimal number of constituents, namely Dhh1, RNA, ATP and Pat1 higher-order dynamic liquid droplets can be formed *in vitro*. The *in vitro* droplets recapitulate various properties of PBs formed *in vivo* such as (a) the stoichiometry of PB components, and (b) the dynamics of Pat1-stimulated assembly and Not1-stimulated disassembly of *in vivo* PBs identifying Pat1 as an important player in PB formation promoting LLPS via Dhh1.

## Discussion

### Tug-of-war model between Pat1 and Not1 to regulate PB dynamics

The ability of proteins and RNA to undergo LLPS has emerged as an essential fundamental biological process allowing cells to organize and concentrate their cellular components without the use of membranes. Yet, how these membraneless organelles assemble, are kept dynamic and are regulated remains enigmatic. One such cellular phase-separated compartment is the PB and our results presented here together with our prior work (*Mugler et al., 2016*) identify and characterize both positive and negative regulators of PB assembly and provides novel insight into the regulation of membraneless organelle formation.

Overall, our work uncovers a function of Pat1 as an enhancer of PB formation that acts via Dhh1 and counteracts the inhibitory function of Not1. Based on our results, we propose that there are at least three inputs that cooperatively regulate PB dynamics *in vivo* (*Figure 7C*). First, the availability of mRNA clients acting as seeding substrates to initiate PB formation. mRNA availability is inversely proportional to the translation status, which is regulated by different stress responses and the metabolic state of the cell (*Figure 2C*). Second, the activity of cAMP-dependent PKA negatively regulates PB assembly in nutrient-rich conditions, at least in part through phosphorylation of Pat1. However, since expression of non-phosphorylatable Pat1$^{AA}$ at endogenous levels (in contrast to overexpression) is not sufficient to induce PBs, Pat1 availability appears to be limiting (*Figures 5* and *4A*, and *Figure 4—figure supplement 1A* and *Ramachandran et al., 2011*). Third, activation of the Dhh1 ATPase cycle by the CCR4-Not1 deadenylation complex stimulates mRNA release and negatively regulates PB assembly. It is of note that Not1 itself might also be subject to post-translational regulation, in response to metabolic state and/or cell cycle stages of the cell (*Mugler et al., 2016*; *Braun et al., 2014*).

In this model, a tug-of-war between Pat1 and Not1 regulates PB assembly and disassembly: whereas a tight Pat1-Dhh1 interaction in the presence of RNA clients, promotes PB formation, Not1, induces PB turnover via stimulating the ATPase activity of the DEAD box ATPase Dhh1 (*Figure 7C*). Remarkably, our *in vivo* findings are corroborated *in vitro* wherein, Pat1 enhances the LLPS of Dhh1 and RNA, a step critical for the assembly of large mRNP granules, and by contrast, Not1 reverses

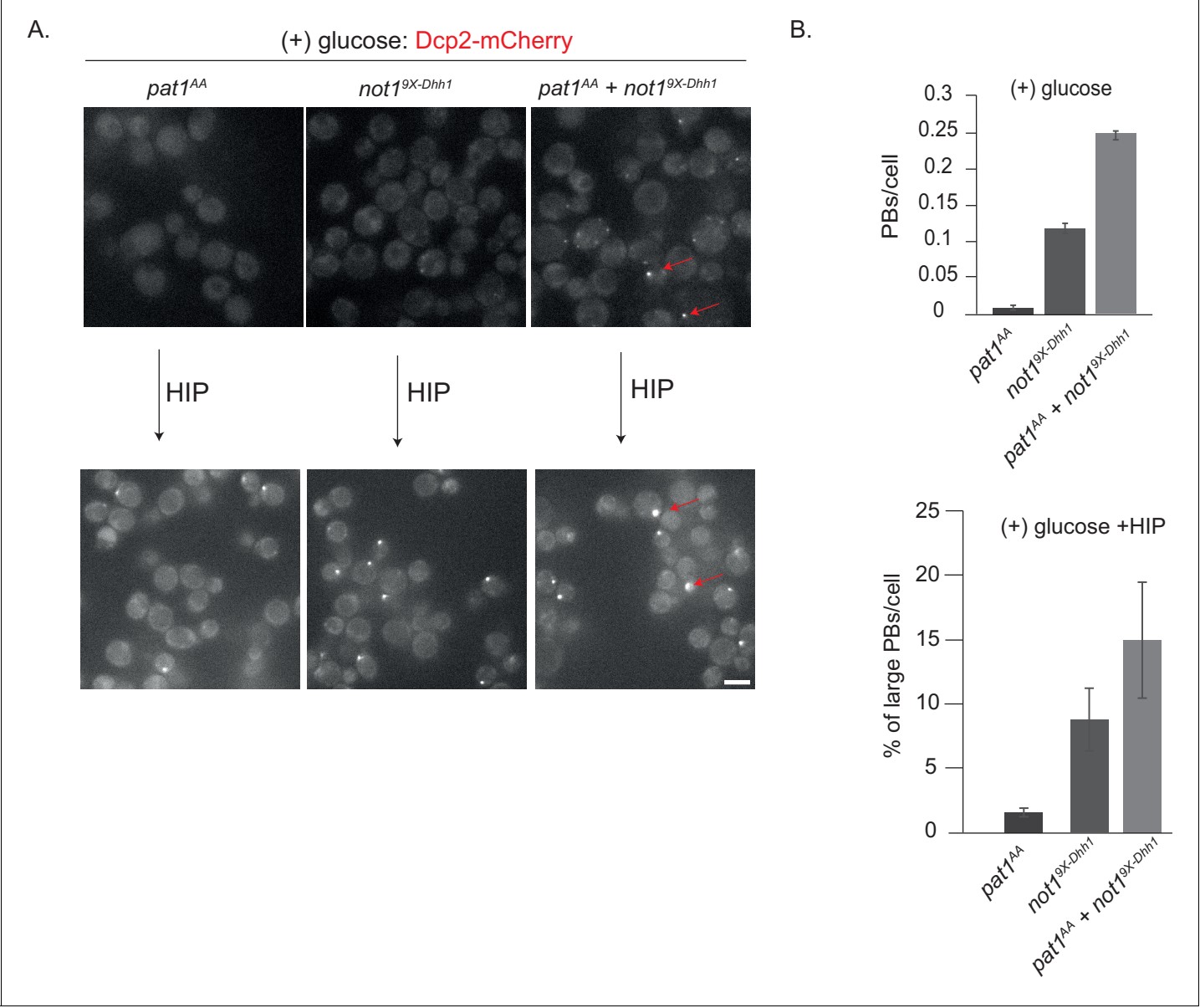

**Figure 5.** *not1^{9X-Dhh1}* (Dhh1 binding mutant) combined with *pat1^{AA}* (non-phosphorylatable Pat1) enhances constitutive PB formation in the absence of stress. (**A**) Cells expressing either *not1^{9X-Dhh1}* or *pat1^{AA}* or the combination of both mutants were grown in SCD media to exponential growth phase and PBs visualized by fluorescence microscopy using Dcp2-mCherry as a *bona fide* PB marker. Scale bar: 5 µm. (**B**) The graph depicts the number of Dcp2 positive PBs/cell. N = 3 biological replicates with >500 cells visualized in each replicate, bars: SEM.
DOI: https://doi.org/10.7554/eLife.41415.016

the LLPS of Pat1-Dhh1-RNA droplets (*Figures 6* and *7C*). Furthermore, the observed stoichiometry of Dhh1 and Pat1 in PBs is recapitulated in our *in vitro* phase separation assay (*Figure 7A,B*).

## The interaction between Pat1 and Dhh1 is critical for PB assembly

Inducing PB formation by Pat1 overexpression or hippuristanol treatment allowed us to dissect distinct mechanisms that govern PB assembly and characterize the crucial role of the Pat1-Dhh1 interaction in this process. The benefit of these modes of PB formation are that they bypass the characteristic stresses associated with PB formation, such as nutrient starvation, that might have more widespread and confounding effects on translation, mRNA degradation, or on the regulation of other PB components.

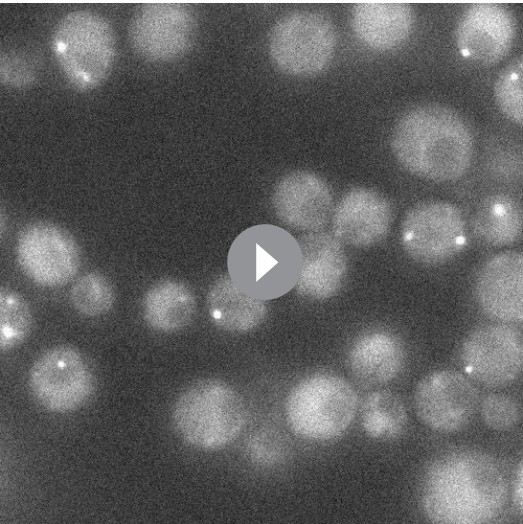 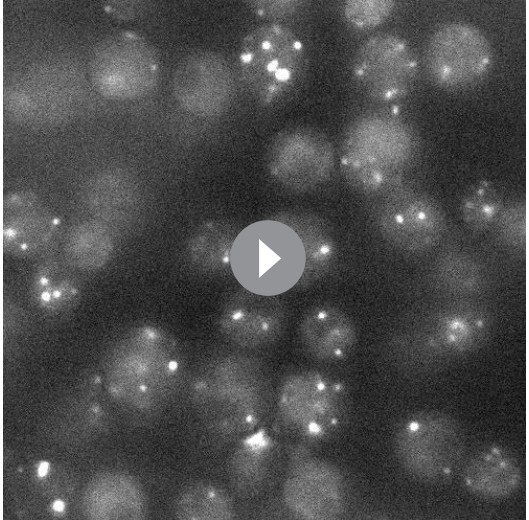

**Video 7.** Cycloheximide treatment causes wild-type PB disassembly. Dcp2-mCherry expressing cells in the *DHH1* background were grown in (SC) raffinose media to exponential growth phase, after which galactose was added to 2% final concentration for 2 hr and then starved for 30 mins to allow PBs to form. The cells were then treated with 50 µg/mL cycloheximide for 90 min and disappearance of foci was monitored using fluorescence microscopy. (5 min intervals; video played at seven fps). Each frame is a single plane.
DOI: https://doi.org/10.7554/eLife.41415.017

**Video 8.** Cycloheximide treatment causes Pat1 OE PB disassembly albeit faster than PB disassembly in cells expressing *dhh1*$^{DQAD}$. Cells expressing Dcp2-mCherry in the *p(GAL)-PAT1* OE strain were grown in (SC) raffinose to exponential growth phase, after which galactose was added to 2% final concentration for 2 hr to allow PBs to form. Thereafter, the cells were treated with 50 µg/mL cycloheximide (CHX) for 90 min and disappearance of foci was monitored using fluorescence microscopy. (5 min intervals; video played at seven fps). Each frame is a single plane.
DOI: https://doi.org/10.7554/eLife.41415.018

Our experiments reveal that Pat1 functions in PB formation through Dhh1 since PB induction by Pat1 overexpression is strictly dependent on the presence of Dhh1 (*Figure 1A*). Furthermore, a mutant in the DETF motif in the N-terminus of Pat1 mediating direct binding to Dhh1 (Pat1$^{4A-Dhh1}$ variant) and mutants of Dhh1 that abolish Pat1 binding fail to promote PB assembly (*Figure 2*). These results were further confirmed using the drug hippuristanol that inhibits translation initiation wherein PBs were induced in wild-type cells but not in cells expressing the Pat1$^{4A-Dhh1}$ mutant, and the Dhh1 mutants that abolish Pat1 binding (Dhh1$^{R295D}$ and Dhh1 $^{S292DN294D}$) [*Figures 2F* and *3C* and *Chan et al., 2018*]. This suggests that high mRNA load alone is inadequate for PB formation and that a direct physical interaction between Pat1 and Dhh1 is obligatory as well. Since the DETF motif in Pat1 is conserved across evolution and was suggested to regulate PB assembly in human cells as well (*Sharif et al., 2013*; *Ozgur and Stoecklin, 2013*), the Pat1-Dhh1 interaction likely plays a critical role in PB formation across species.

Our results seem to be at odds with a previous publication, which suggests that Pat1 acts independently of Dhh1 to promote PB formation (*Coller and Parker, 2005*). However, in this study Pat1 was overexpressed from a plasmid, in the presence of the endogenous copy. When recapitulating these conditions, we observed that PB components mislocalize to the nucleus [observed by DAPI and a nuclear rim marker, see *Figure 7—figure supplement 2A,B*] and it is possible that in the absence of additional organelle markers, this nuclear accumulation was misinterpreted as PB formation.

## Pat1 enhances the LLPS of Dhh1, thereby promoting PB formation

How does Pat1 enhance PB formation? One hypothesis is that Pat1 directly counteracts the Not1-stimulated ATPase activation of Dhh1, but we do not favor such a model since i) Pat1 does not inhibit Dhh1's ATPase activity at physiological protein ratios *in vitro* (*Figure 6—figure supplement 1A*), ii) Not1$^{MIF4G}$ dissolves Pat1-Dhh1 condensates, while droplets formed from catalytic-dead Dhh1$^{DQAD}$ are Not1-resistant (*Figure 6* and *Mugler et al., 2016*), and iii) *in vivo*, PBs formed upon

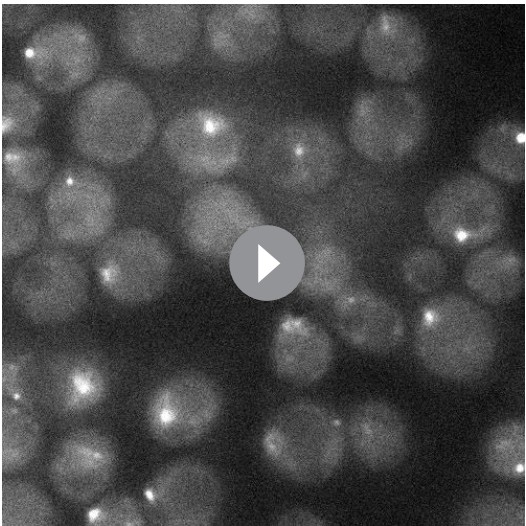

**Video 9.** Cycloheximide treatment of *dhh1* $^{DQAD}$ PBs. Dcp2-mCherry expressing cells in the *dhh1*$^{DQAD}$ mutant background were grown in (SC) raffinose to exponential growth phase, after which galactose was added to 2% final concentration for 2 hr. The cells were then carbon starved for 30 min to allow PBs to form after which, 50 μg/mL cycloheximide was added and disappearance of foci was monitored for 90 mins using fluorescence microscopy. (5 min intervals; video played at seven fps). Each frame is a single plane.
DOI: https://doi.org/10.7554/eLife.41415.019

Pat1 overexpression are much more dynamic than those formed from Dhh1$^{DQAD}$ (*Figure 7—figure supplement 1B*).

Instead, our data are consistent with a model in which Pat1 promotes PB assembly by directly enhancing the LLPS of Dhh1. DDX6, the human homolog of Dhh1 can oligomerize (*Ernoult-Lange et al., 2012*), and given the fact that both DDX6 and Dhh1 are found in molar excess over mRNA, we propose that Pat1 functions as a scaffold, providing additional multivalent interactions with Dhh1 and RNA, thereby aiding in the fusion of Dhh1 LLPS droplets and promoting assembly of microscopically-detectable mRNP condensates (*Figure 6*). Interestingly, both functional binding to Dhh1 in Pat1-N and RNA-binding in Pat1-C are required for this mechanism as individually mutations in each of the domains alter PB assembly *in vivo* and fail to enhance Dhh1's phase separation *in vitro*. Rather, Pat1-N alone diminishes the extent of LLPS (*Figure 6—figure supplement 2C*). It is interesting to note that Pat1-N was previously shown to interfere with the ability of Dhh1 to interact with RNA (*Sharif et al., 2013*) and we thus speculate that Pat1-N might outcompete RNA from Dhh1, and in the absence of the C-terminal Pat1 RNA-binding site destabilize droplets. This might explain why RNA-binding by Pat1's C-terminus that is under the regulation of PKA (*Figure 4A,B*, *Figure 4—figure supplement 1A*) is critical for LLPS and PB assembly. *In vivo*, we find that the Pat1$^{4A-Dhh1}$ (Dhh1 binding mutant) and Pat1$^{EE}$ (phospho-mimetic mutant) not only have a drastically reduced number of PBs, but also much lower intensity (reduced by 7 and 8 fold respectively, *Figure 7—figure supplement 2C*) compared to Pat1 wild-type PBs.

Our recent findings also show that the un-structured poly-Q rich C-terminal tail of Dhh1 is required for PB formation *in vivo* and Dhh1 LLPS *in vitro* (Hondele et al., *submitted*). Thus, Pat1 might potentially enhance Dhh1 multimerization via the low complexity tails of Dhh1. Structural studies will now be needed to clarify how Pat1 facilitates oligomerization and PB formation of the Dhh1-Pat1- RNA complex.

## The emerging importance of LLPS in diverse aspects of cell biology and the power of *in vitro* reconstitution systems

As non-membrane bound organelles, it is of interest to understand how cells assemble large RNP granules and regulate their dynamics. PBs are prominent membraneless cellular compartments, which have been shown to be not only involved in numerous aspects of mRNA turnover, but also crucial for survival under various cellular stresses. With an *in vitro* phase separation tool in hand that faithfully reconstitutes certain aspects of PB formation, we are now in a position to dissect how PBs form and characterize their biophysical behavior. This will aid us not only in elucidating key components of PB formation, but also help us elucidate the function of PBs in regulating mRNA turnover.

At this stage, our *in vitro* systems are limited to Dhh1, Pat1, Not1 and RNA and for the sake of simplicity; we present a model only depicting the role of Pat1 and Dhh1 in PB formation (*Figure 7C*). However, it is important to note that several additional PB components have been characterized. For example, this includes Ecd3, members of the Lsm1-7 complex, or Dcp2 that play crucial roles either alone or together with Pat1 and Dhh1 in PB assembly. Both Edc3 and Lsm4 contain low complexity-domains that were shown to be critical for PB formation *in vivo*, and it was recently demonstrated a threshold concentration of low-complexity domain containing proteins is required

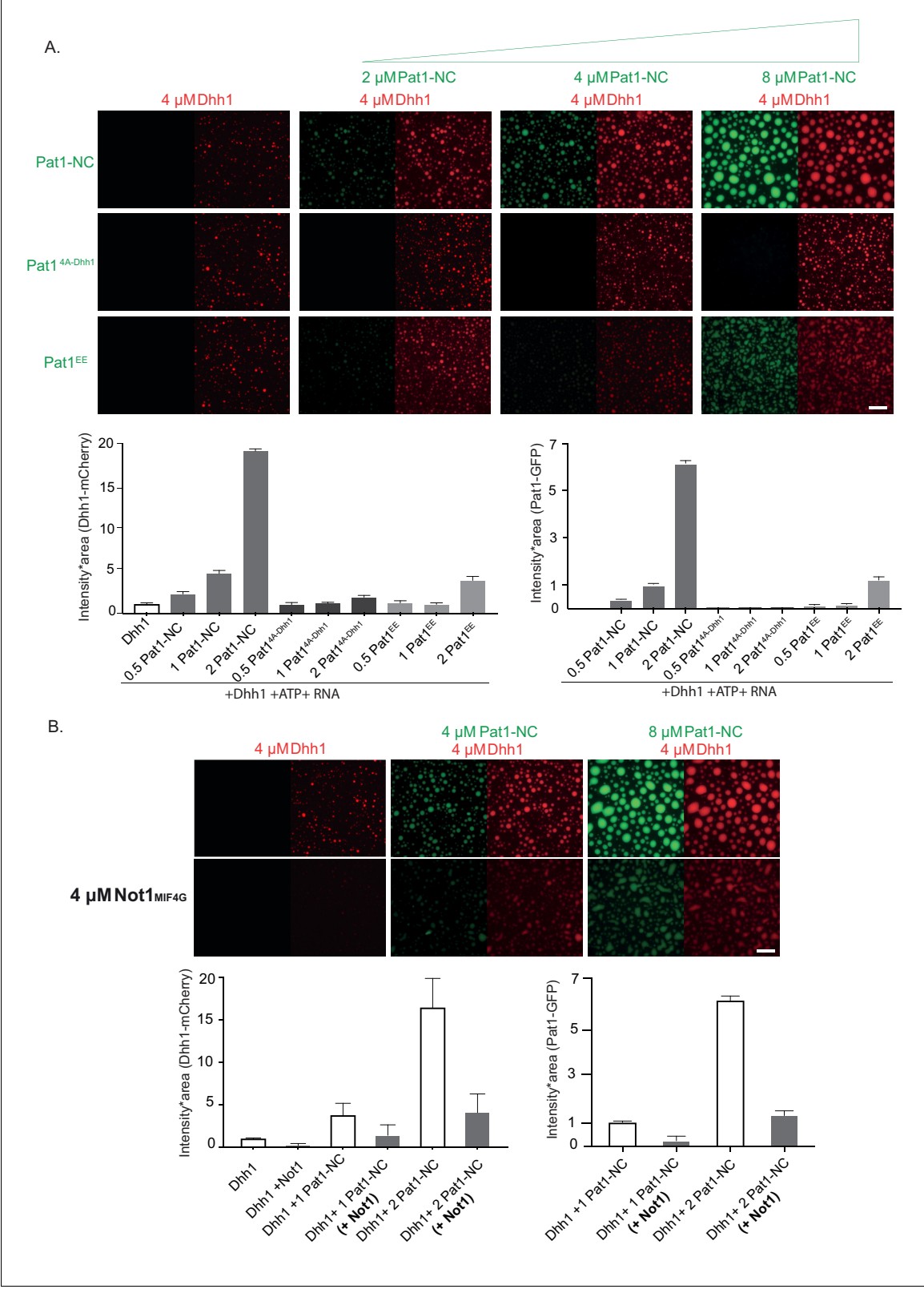

**Figure 6.** Pat1 WT but not Pat1[4A-Dhh1/EE] mutant enhances the phase separation of Dhh1 and RNA *in vitro*. (**A**) 4 µM Dhh1-mCherry was mixed with 3.2 mM ATP and 0.2 mg/ml polyU as RNA-analog in a 150 mM KCl buffer. Pat1-NC-GFP and mutants thereof were added in increasing concentrations, as indicated. mCherry and GFP intensities are scaled identically in all images displayed. Representative images, scale bar 25 µm. Quantification of the droplet intensity multiplied by area (mCherry and GFP channels separately normalized to Dhh1 alone and Pat1 4 µM, respectively) of three technical

*Figure 6 continued on next page*

*Figure 6 continued*

replicates of the reactions displayed in *Figure 5A*, mean and standard deviation. (B) Not1 prevents droplet formation also in the presence of Pat1. Proteins were mixed as indicated, and droplet formation induced by addition of low salt buffer, ATP and polyU. Scale bar 25 mM, quantification (intensity multiplied by area) of two biological replicates (one to three technical replicates each), mean and standard deviation.

DOI: https://doi.org/10.7554/eLife.41415.020

The following figure supplements are available for figure 6:

**Figure supplement 1.** Pat1 does not slow down the ATPase cycle of Dhh1.

DOI: https://doi.org/10.7554/eLife.41415.021

**Figure supplement 2.** Pat1 does not phase separate *in vitro* and the enhancement of Dhh1's phase separation via Pat1 is RNA dependent.

DOI: https://doi.org/10.7554/eLife.41415.022

for robust PB assembly (*Protter et al., 2018*). In addition, it was previously shown that both Pat1 (C-terminus) and the Lsm1-7 complex individually have low RNA binding ability but a reconstituted Lsm1-7-Pat1-C complex displays an enhanced ability to bind RNA (*Chowdhury et al., 2014*). This might suggest that *in vivo* the Pat1-C- terminus functions together with the Lsm1-7 complex to bind RNA and enhance PB formation. Furthermore, it cannot be excluded that direct binding of Dcp2 to the C-terminus of Pat1 also regulates PB assembly and RNA turnover (*Fourati et al., 2014*). It will thus be critical to examine additional PB components in future experiments and to evaluate how they function together with Pat1 and Dhh1 in order to understand the complex interplay of various inputs that regulate PB assembly and dynamics. Nonetheless, based on our data and the aforementioned literature evidence, we propose that Pat1 provides a central PB scaffold and participates in an intricate network of protein–protein and protein-RNA interactions that promote PB assembly and influence RNA turnover rates *in vivo*.

The physical basis of LLPS has attracted a great deal of attention recently, at least in part because of the critical role that proper mRNP assembly plays in pathological neurodegenerative diseases and stress responses (*Ramaswami et al., 2013*; *Alberti and Hyman, 2016*; *Patel et al., 2015*; *Aguzzi and Altmeyer, 2016*; *Hyman et al., 2014*). An important challenge is to understand how these granules are kept dynamic and functional under some conditions whereas the formation of solidified and aberrant aggregates are triggered under others. Robust *in vitro* reconstitution systems that can recapitulate mRNP granule assembly, dynamics and stoichiometry will be invaluable in addressing these research questions and increase our understanding of the molecular mechanism of how cells age and deal with stress.

# Materials and methods

**Key resources table**

| Reagent type (species) or resource | Designation | Source or reference | Identifiers | Additional information |
|---|---|---|---|---|
| Strain, strain background (Yeast: *Saccharomyces cerevisiae*) | W303 | SGD:https://www.yeastgenome.org/strain/S000203491 | KWY XYZ | MATa/MATα {leu2-3,112 trp1-1 can1-100 ura3-1 ade2-1 his3-11,15} [phi+] |
| Strain, strain background (*Escherichia coli*) | *E.Coli* DH5α | Thermo Fisher Scientific | 18258012 | |
| Strain, strain background (*Escherichia coli*) | *E.Coli* BL21 star (DE3) | Thermo Fisher Scientific | C601003 | |
| Genetic reagent () | | | | Please see *Supplementary file 1, 2*—S2A |
| Antibody | mouse-anti-GFP | Roche | Cat# 11814460001, RRID: AB_390913 | Western blot: 1:1000 |

*Continued on next page*

*Continued*

| Reagent type (species) or resource | Designation | Source or reference | Identifiers | Additional information |
|---|---|---|---|---|
| Antibody | rabbit-anti-Hxk1 | US Biological | Cat# H2035-01, RRID: AB_2629457, Salem, MA | Western blot: 1:3000 |
| Antibody | IRdye 680RD goat-anti-rabbit | LI-COR Biosciences | Cat# 926–68071, RRID: AB_10956166 | Western blot: 1:5000 |
| Antibody | IRdye 800 donkey-anti-mouse | LI-COR Biosciences | Cat# 926–32212, RRID: AB_621847 | Western blot: 1:10000 |
| Antibody | Rabbit-anti-Dhh1 | in house | #100 | Western blot: 1:5000 |
| Recombinant DNA reagent | | | | Please see *Supplementary file 2*—table S2B |
| Chemical compound, drug | hippuristanol | kind gift from Junichi Tanaka, University of the Ryukyus | hippuristanol | 10 µM in DMSO |
| Chemical compound, drug | cyclohexi | Sigma-Aldrich, CH | CAS Number 66-81-9 | 50 ug/ml in DMSO |
| Software, algorithm | Diatrack | http://www.diatrack.org/ | Diatrack | used for counting PB number and intensity |
| Software | fiji/imagej | NIH | https://fiji.sc/ | adjusting brightness, contrast and making final figures |

The details regarding all the yeast strains used in this study are in *Supplementary file 1*. *Supplementary file 2*—table S2A contains all the plasmids used in this study. Table Supplementary S2B contains the mutants used in this study and *Supplementary file 2*—table S2C describes the entire DNA oligos used for this manuscript.

## Construction of yeast strains and plasmids

*S. cerevisiae* strains used in this study are derivatives of W303 and are described in *Supplementary file 1*. ORF deletion strains and C-terminal epitope tagging of ORFs was done by PCR-based homologous recombination, as previously described (*Longtine et al., 1998*). Plasmids for this study are described in *Supplementary file 2*—table S2A. Mutations in Pat1 were generated by introducing the mutation in the primer used to amplify the respective Pat1 regions and stitched together with the selection marker from the plasmid (*Supplementary file 2*—table S2B). Mutations in Dhh1 and Not1 were generated as in *Mugler et al. (2016)*. Primer sequences for strain construction are listed in table S2C.

## Overexpression of Pat1 wild-type and mutants

The samples were grown overnight in synthetic media containing 2% raffinose, diluted to $OD_{600} = 0.05$ or 0.1 the following day, and grown to mid-log phase ($OD_{600} = 0.3–0.4$). The culture was split into two and to one-half galactose was added to 2% final concentration and the corresponding protein induced for 2–3 hr. The cells in both raffinose and galactose were imaged using a wide-field fluorescence microscope.

## PB induction and disassembly kinetics

PBs were induced via glucose starvation stress. Samples were grown overnight in synthetic media containing 2% glucose, diluted to $OD_{600} = 0.05$ or 0.1 the following day, and grown to mid-log phase ($OD_{600} = 0.3–0.8$). Cells were harvested by centrifugation and washed in ¼ volume of fresh

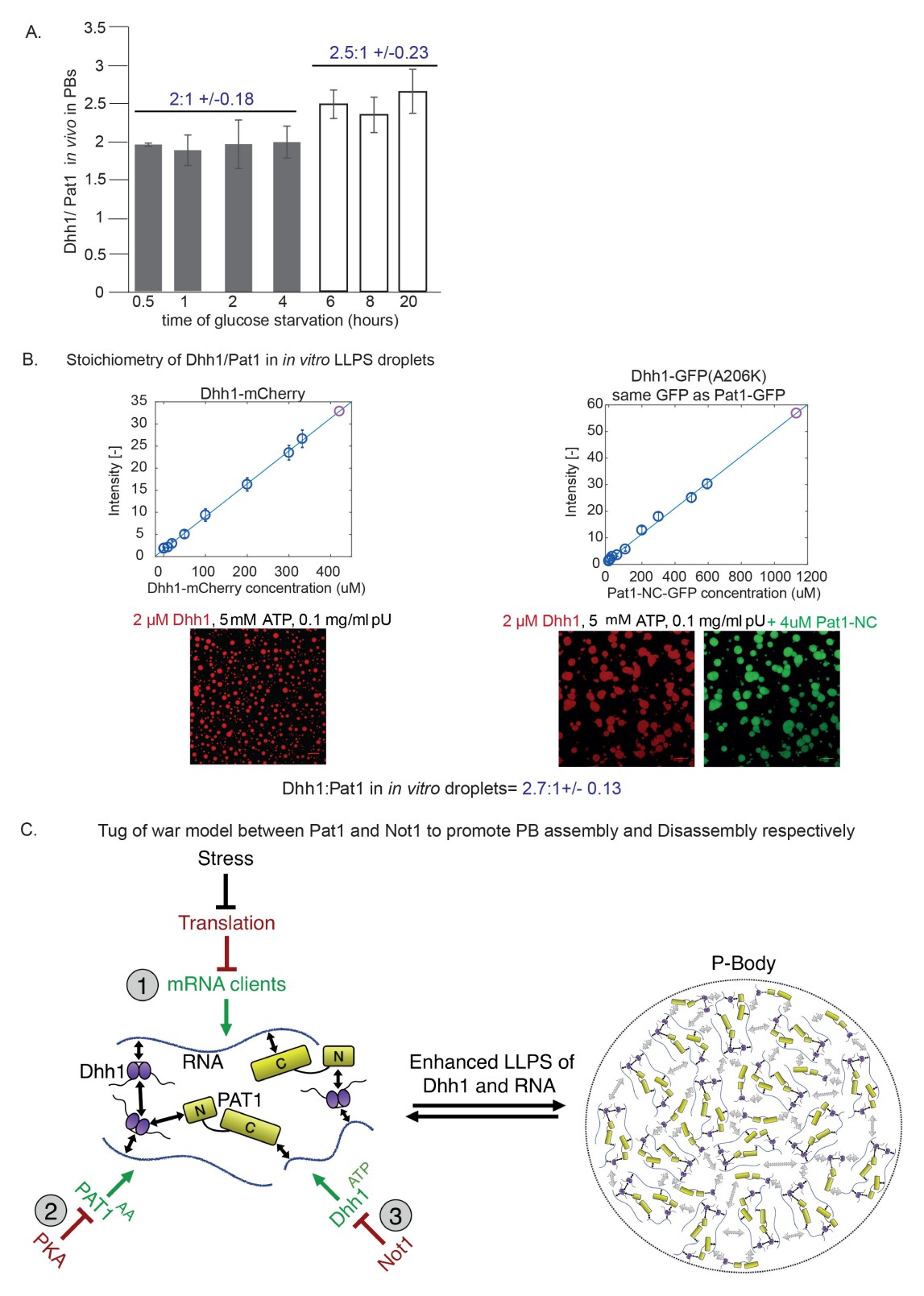

**Figure 7.** Stoichiometry of Pat1 and Dhh1 in PBs *in vivo* and in liquid droplets *in vitro*. (**A**) Pat1-GFP and Dhh1-GFP expressing cells were grown in SCD media to exponential growth phase and then shifted to SC (minus glucose) for the indicated time points. PB formation was visualized by fluorescence microscopy. Graph depicts the ratio of Dhh1: Pat1 in PBs *in vivo*. N = 3 biological replicates, SEM. (**B**) Calibration curves to determine the protein concentration of Dhh1-mCherry and Pat1-NC-GFP within the droplets: the fluorescence intensity of homogeneous solutions of the respective proteins

*Figure 7 continued on next page*

*Figure 7 continued*

was measured at different protein concentrations (blue circles). The unknown protein concentration was determined by linear fitting of this calibration curve (violet circle). Confocal fluorescence microscopy images of phase-separated droplets of Dhh1-mCherry in the absence and presence of Pat1-GFP are shown. Scale bar: 20 μM. (C) Tug-of-war model between Pat1 and Not1 to regulate PB dynamics. Model depicting three inputs that cooperatively regulate PB formation. The solid black arrows on the left show interaction between the Pat1 N-terminus with the RecA2 core of Dhh1, Pat1 C-terminus with RNA, Dhh1 interaction with RNA and potential Dhh1-Dhh1 interactions via low-complexity unstructured tails. The dotted grey arrows on the right demonstrate multivalent interactions in a PB driven by the LLPS of Dhh1 and RNA (a process enhanced by Pat1). Colors: green: inputs that promote PB assembly, red factors that negatively regulate PB formation. Violet: Dhh1, light green: Pat1-NC, blue: RNA.

DOI: https://doi.org/10.7554/eLife.41415.023

The following figure supplements are available for figure 7:

**Figure supplement 1.** Images and protein expression levels related to stoichiometry of Pat1 and Dhh1 in PBs *in vivo* and in liquid droplets *in vitro*.

DOI: https://doi.org/10.7554/eLife.41415.024

**Figure supplement 2.** OE of Pat1 in the presence of the endogenous copy of Pat1 [*p-PAT1* +p(GAL)-*PAT1*] leads to the nuclear localization of PB components.

DOI: https://doi.org/10.7554/eLife.41415.025

synthetic media ± 2% glucose, then harvested again and re-suspended in 1 vol of fresh synthetic media ± 2% glucose and grown 30 min at 30°C. Cells were then transferred onto Concanavalin A-treated MatTek dishes (MatTek Corp., Ashland, MA) and visualized at room temperature. For cycloheximide experiments, final concentration of 50 ug/mL (Sigma-Aldrich, CH) were used to examine PB disassembly kinetics. For PB induction and to determine the kinetics of PB assembly we used hippuristanol (a generous gift of Junichi Tanaka, University of the Ryukyus) at a final concentration of 10 μM.

## Wide-field fluorescence microscopy

Cells were transferred onto Concanavalin A-treated MatTek dishes (MatTek Corp., Ashland, MA) and visualized at room temperature Microscopy was performed using an inverted epi-fluorescence microscope (Nikon Ti) equipped with a Spectra X LED light source and a Hamamatsu Flash 4.0 sCMOS camera using a 100x Plan-Apo objective NA 1.4 and the NIS Elements software. Representative images were processed using ImageJ software. Brightness and contrast were adjusted to the same values for images belonging to the same experiment and were chosen to cover the whole range of signal intensities. Image processing for PB analysis was performed using Diatrack 3.05 particle tracking software (*Vallotton and Olivier, 2013*) as described below.

## Automated image analysis for PB quantification

In order to quantify PB formation in live cells, we used an automated image analysis in a manner similar to *Mugler et al. (2016)*. First, PBs were counted using Diatrack 3.05 particle tracking software using local intensity maxima detection, followed by particle selection by intensity thresholding and particle selection by contrast thresholding with a value of 5% (*Vallotton et al., 2017*). To speed up the analysis, we renamed all our images in a form that can be recognized as a time-lapse sequence by Diatrack, and placed them all in a single directory, such that they all will be analyzed using exactly the same image analysis parameters. Renaming and copying was done by a custom script (*Source Code 1*), which also performed cell segmentation using a method adapted from (*Hadjidemetriou et al., 2008*). Briefly, the method first detects all edges using a Laplacian edge detection step, and then traces normals to those edges in a systematic manner. These normals tend to meet at the cell centre where the high density of normals is detected, serving as seeds to reconstruct genuine cells. Our script thus counts cells and reports their number for each image - information which is output to an excel table. The results from Diatrack PB counting are imported from a text file into that table, and the number of PB is divided by the number of cells for each image.

## Protein purification

Dhh1 and Not1 were purified as described previously (*Mugler et al., 2016*). V5-Pat1-C, GST-Pat1-N and GST-Pat1-NC constructs as described in *Supplementary file 2*—table S2A were cloned into pETM-CN vectors. These expression vectors were transformed into chemically competent *E. coli* BL21 DE3 under the selection of ampicillin and chloramphenicol. Pre-cultures were grown in LB at

37°C over-night, and diluted 1:100 into rich medium the next morning. Cells were grown at 37°C to an OD600 of 0.6 and induced with 200 mM IPTG (final concentration). Cells were then grown over-night at 18°C, harvested and resuspended in 30 mL lysis buffer (500 mM (Pat1-C, Pat1-N) or 300 mM NaCl (Pat1-NC), 25 mM Tris-HCl pH 7.5, 10 mM imidazole, protease inhibitors, 10% glycerol) per cell pellet from 2 L of culture. After cell lysis by EmulsiFlex (Avestin Inc, Ottawa, CA), the 6xHis tagged proteins were affinity extracted with Ni$^{2+}$ sepharose in small columns, dialyzed into storage buffer (MH200G (Pat1-C, Pat1-N): 200 mM NaCl, 25 mM Tris pH 7.5 (RT), 10% glycerol, 2 mM DTT), MH300G (Pat1-NC): same with 300 mM NaCl) with simultaneous protease cleavage of the His-tag and GST-tag (unless required for pull-down assays) and further purified by size exclusion with a Superdex 200 column on an AEKTA purifier (both GE Life Sciences, Marlborough, MA) in storage buffer. Protein expression levels, His eluates and gel filtration fractions were analyzed by SDS-PAGE. Clean Superdex elution fractions were pooled, concentrated using Millipore Amicon Centrifugation units and snap frozen as ~20 µl aliquots in siliconized tubes in liquid nitrogen.

### ATPase assays

ATPase assays were performed according as described (*Montpetit et al., 2011*) with the following modifications: final concentration 2 µM Dhh1-mCherry was mixed with 0.5 µM Not1 and Pat1-mCherry (1 and 3 µM) as indicated and protein volumes equalized with storage buffer (MH200G). 2 µL 10x ATPase buffer (300 mM HEPES-KOH pH 7.5, 1 M NaCl, 20 mM MgCl$_2$), 4 µL 10 mg/ml polyU (unless indicated otherwise), RNase inhibitors, 13.3 µL 60% glycerol, 2.7 µL 10 mg/mL BSA, were added to a final volume of 36 µL. Reactions were set up in triplicate in a 96-well NUNC plate. The assay was initiated by the addition of 40 µL of a master mix containing 1x ATPase buffer, 2.5 mM ATP (from a 100 mM stock in 0.5 M HEPES-KOH pH 7.5), 1 mM DTT, 6 mM phosphoenolpyruvate, 1.2 mM NADH (from a 12 mM stock in 25 mM HEPES-KOH pH 7.5) and 125–250 units/mL PK/LDH. NADH absorption was monitored with a CLARIOstar plate reader (BMG Labtech, Ortenberg, Germany) at 340 nm in 30 s intervals for 400 cycles.

### *In vitro* liquid droplet reconstitution assay

Reactions were pipetted in 384-well microscopy plates (Brooks 384 well ClearBottom Matriplate, low glass). Proteins were diluted to 50 µM (Dhh1) or 100 µM (Not1, Pat1) stocks with storage buffer and mixed as droplets at the side of the well; volumes were equalized to 5 µl with storage buffer. Next, a master mix of 12.5 µl 150 mM KCl buffer (150 mM KCl, 30 mM HEPES-KOH pH 7.4, 2 mM MgCl$_2$), 2 µl ATP reconstitution mix (40 mM ATP, 40 mM MgCl$_2$, 200 mM creatine phosphate, 70 U/ mL creatine kinase), 2.5 µl 2 mg/ml polyU (in H$_2$O) as an RNA analog, 1 µl 1M HEPES-KOH pH 6.4 buffer and 2 ml 10 mg/ml BSA were added and mixed by pipetting, the droplets still at the side of the well, since this later on produced more equal droplet distribution. First imaging was performed after 20 min incubation at room temperature; just prior to imaging, droplets were spun down at 100 g for 1 min. For subsequent analysis, plates were stored in the fridge. Pictures displayed were recorded after 1 hr incubation.

### Stoichiometry analysis of Dhh1 and Pat1 in PBs *in vivo*

In order to estimate the relative amount of Dhh1 to that of Pat1 in PBs *in vivo*, we labelled both proteins with eYGFP in two separate strains, and imaged them in exactly the same conditions after inducing carbon (glucose) starvation stress for indicated times (*Figure 6A*). Using exactly the same endogenous probe for both proteins allowed us to deduce ratios of abundances directly from ratios of intensities (no spectral corrections are necessary). We then measured the intensity of the 10 brightest PBs as a function of time and in both cases took the median intensity value as representative of the more visible PBs (at least 10 PBs were present in every image for both strains and at each time point). Finally, we divided the median value obtained for Dhh1 by that for Pat1 and plotted that ratio as a function of time post-stress.

### Stoichiometry of Dhh1 and Pat1 in phase-separated *in vitro* droplets

For the analysis of the *in vitro* phase transition of Dhh1 in the presence of Pat1, 2 µM mCherry-tagged Dhh1 and 4 µM GFP-tagged Pat1-NC were added to 0.1 mg/ml polyU and 5 mM ATP in 150 mM KCl, 30 mM HEPES-KOH, pH 7.4 and 2 mM MgCl$_2$. After 1 hr incubation, the phase-separated

droplets were imaged in both the mCherry and the GFP channel by confocal fluorescence microscopy (Leica TCS SP8) with a 63x NA 1.4 oil objective (Leica). The protein concentration was evaluated from the fluorescence intensity via a standard curve obtained by measuring a series of samples at different known concentrations for each fluorophore. The size distributions were obtained by analyzing the images obtained by fluorescence microscopy with an in-house code written in Matlab.

### GST pull-down assays

25 µl of 100 µM recombinant purified GST Pat1-NC was bound to 30 µl GST-slurry (Sigma), and one aliquot of GST-beads was included without added Pat1-NC as negative control. Reactions were incubated for 30mins rotating at 4°C and washed 4-times with wash buffer (300 mM NaCl, 10 mM Tris pH 7.5, 0.05 % NP40). 4 µl 250 µM Dhh1 was added in 500 µl wash buffer and incubated rotating at 4°C for 1 hr. Reactions were washed 5-times, proteins eluted by boiling with 4x SDS loading buffer and separated on a 12% acrylamide gel and stained with Coomassie. For V5-pulldown assays, the bait (15 µl 100 µM Pat1-C construct) was bound to 15 ml V5-slurry (Sigma), the bait 3 µl 200 µM Dhh1-mCherry, and incubation/wash buffers as above with salt concentration as indicated in the blot. Final washes were only performed three times.

### Native PAGE

Proteins were diluted to 20 µM. 0, 2 or 4 µl of protein solution was filled to 4 µl with storage buffer (200 mM NaCl, 25 mM Tris pH 7.5, 10% glycerol, 2 mM DTT). Proteins were mixed with a master mix containing per reaction 2 µl 10x ATPase buffer (300 mM HEPES-KOH pH 7.5, 1 M NaCl, 20 mM $MgCl_2$), 0.3 µl Cy5-labeled in vitro transcribed RNA oligo (29nt, 1.1 µg/µl), 2 µl 2.5M NaCl, 13.7 µl $H_2O$ and 3 µl 60% glycerol. Reactions were incubated on ice for 20 min and separated on a precast Bis-Tris Native gradient Gel (Invitrogen) in 1x running buffer (50 mM Bis-Tris 50 mM Tricine pH 6.8) at 100V 4°C for about 2.5 hr.

### Growth curve and spotting assays

Overnight yeast cultures were grown in permissive conditions in SCD media. Growth curves were acquired using CLARIOstar automated plate reader (BMG Labtech, Ortenberg, Germany) at 30°C in 24-well plastic dishes (Thermo Fisher) from overnight pre-cultures diluted 1:125 in the specified synthetic liquid media. For spotting assays on plates the indicated strains were grown overnight at 25°C to saturation. Next day equal number of ODs for each strain was spotted (with a 1:10 series dilution) on YPD plates and incubated at 25°C, 30°C and 37°C for 2–3 overnights.

### Western blotting

For Western blot analysis, roughly 5 $OD_{600}$ units of log growing cells were harvested and treated with 0.1M NaOH for 15 mins at room temperature. The samples were vortexed in between and finally centrifuged to remove the NaOH. SDS sample buffer was added and homogenates were boiled. Proteins were resolved by 4–12% Bolt Bis-Tris SDS PAGE (Thermo Fischer, Waltham, MA), then transferred to nitrocellulose membrane (GE Life Sciences, Marlborough, MA). Membranes were blocked in PBS with 5% non-fat milk, followed by incubation with primary antibody overnight. Membranes were washed four times with PBS with 0.1% Tween-20 (PBS-T) and incubated with secondary antibody for 45 min. Membranes were imaged and protein bands quantified using an infrared imaging system (Odyssey; LI-COR Biosciences, Lincoln, NE). The following primary antibodies were used for detection of tagged proteins at the indicated dilutions: rabbit-anti-Dhh1 (1:5000) as described in (*Fischer and Weis, 2002*), (Weis Lab ETH Zurich Cat# Weis_001, RRID:AB_2629458), mouse-anti-GFP (1:1000) (Roche Cat# 11814460001, RRID:AB_390913), and rabbit-anti-Hxk1 (1:3000) (US Biological Cat# H2035-01, RRID:AB_2629457, Salem, MA). IRdye 680RD goat-anti-rabbit (LI-COR Biosciences Cat# 926–68071, RRID:AB_10956166) and IRdye 800 donkey-anti-mouse (LI-COR Biosciences Cat# 926–32212, RRID:AB_621847) were used as secondary antibodies.

## Acknowledgements

We would like to thank Justine Kusch and the ETHZ ScopeM facility for technical assistance with confocal microscopy, Elisa Dultz, Carmen Weber, Stephanie Heinrich, Mostafa Zedan, Sarah Khawaja

Evgeny Onishchenko, Annemarie Kralt and Jan Nagler for their critical reading of this manuscript and members of the Weis lab for helpful discussions and comments. We would like to thank Bram Hofland for assisting in cloning of yeast constructs. We would also like to thank the Séraphin lab for sharing their yeast strains. MH was supported by a Human Frontier Science Program (HFSP) post-doctoral fellowship (LT000914/2015) and an ETH postdoctoral fellowship (FEL-37-14-2). This work was supported by NIH/NIGMS (R01GM058065 and R01GM101257 to KW) and the Swiss National Science Foundation (SNF 31003A_159731 and 31003A_179275 to KW).

## Additional information

### Competing interests

Karsten Weis: Reviewing editor, *eLife*. The other authors declare that no competing interests exist.

### Funding

| Funder | Grant reference number | Author |
| --- | --- | --- |
| Human Frontier Science Program | LT000914/2015 | Maria Hondele |
| ETH Zürich | FEL-37-14-2 | Maria Hondele |
| Schweizerischer Nationalfonds zur Förderung der Wissenschaftlichen Forschung | 31003A_159731 | Karsten Weis |
| National Institute of General Medical Sciences | R01GM058065 | Karsten Weis |
| National Institute of General Medical Sciences | R01GM101257 | Karsten Weis |
| Schweizerischer Nationalfonds zur Förderung der Wissenschaftlichen Forschung | 31003A_179275 | Karsten Weis |

The funders had no role in study design, data collection and interpretation, or the decision to submit the work for publication.

### Author contributions

Ruchika Sachdev, Conceptualization, Data curation, Formal analysis, Validation, Investigation, Methodology, Writing—original draft, Writing—review and editing; Maria Hondele, Conceptualization, Formal analysis, Validation, Investigation, Methodology, Writing—review and editing; Miriam Linsenmeier, Conceptualization, Formal analysis, Investigation, Methodology, Writing—review and editing; Pascal Vallotton, Data curation, Formal analysis, Methodology, Writing—review and editing; Christopher F Mugler, Conceptualization, Resources, Writing—review and editing; Paolo Arosio, Conceptualization, Formal analysis, Methodology, Writing—review and editing; Karsten Weis, Conceptualization, Supervision, Funding acquisition, Validation, Writing—original draft, Project administration, Writing—review and editing

### Author ORCIDs

Ruchika Sachdev http://orcid.org/0000-0002-4472-3417
Maria Hondele http://orcid.org/0000-0002-2733-2561
Christopher F Mugler https://orcid.org/0000-0001-8258-1192
Karsten Weis http://orcid.org/0000-0001-7224-925X

### Decision letter and Author response

Decision letter https://doi.org/10.7554/eLife.41415.032
Author response https://doi.org/10.7554/eLife.41415.033

## Additional files

### Supplementary files

• Source code file 1. MATLAB script to count the yeast cell number in an automated manner.
DOI: https://doi.org/10.7554/eLife.41415.026

• Supplementary file 1. Details regarding all the yeast strains used in this study.
DOI: https://doi.org/10.7554/eLife.41415.027

• Supplementary file 2. Details regarding the plasmids , mutants and DNA oligos used in this study. Supplementary table S2A contains all the plasmids used in this study. Table Supplementary S2B contains the mutants used in this study and Table S2C describes the entire DNA oligos used for this manuscript.
DOI: https://doi.org/10.7554/eLife.41415.028

• Transparent reporting form
DOI: https://doi.org/10.7554/eLife.41415.029

### Data availability

All data generated or analysed during this study are included in the manuscript and supporting files.

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
