## [Decision Letter]

Thank you for submitting your article "Pat1 promotes processing body assembly by enhancing the phase separation of the DEAD-box ATPase Dhh1 and RNA" for consideration by *eLife*. Your article has been reviewed by three peer reviewers, including Alan Hinnebusch as the Reviewing Editor, and the evaluation has been overseen by James Manley as the Senior Editor. The following individual involved in review of your submission has agreed to reveal her identity: Martine A Collart.

The reviewers have discussed the reviews with one another and the Reviewing Editor has drafted this decision to help you prepare a revised submission.

Summary:

In this work the authors address the role of an activator of mRNA decapping called Pat1 in the formation and disassembly of P-bodies (PB) in budding yeast. Using both *in vivo* analyses and *in vitro* assays the authors conclude that Pat1 promotes PB formation by interaction with the Dhh1 ATPase, another decapping activator and functions in opposition to Not1, which instead reverses Dhh1-RNA condensation.

The authors show that overexpression of Pat1 leads to PB formation, identify a minimal Pat1 derivative that is sufficient for this and enables PB formation upon glucose depletion, and show that this minimal derivative interacts with Dhh1. They use mutants that disrupt interaction with Dhh1 to determine that the interaction is necessary for PB formation in response to glucose or inhibition of translation initiation. They also provide evidence that phosphorylation of Pat1 regulates its contribution to PB formation, and ability to interact directly with RNA, but not its interaction with Dhh1, by using non-phosphorylatable or phosphomimetic mutants of the phosphorylation site. They further show that the role of Pat1 is not to antagonize the activation of Dhh1 ATPase activity by Not1. They use an *in vitro* system to follow LLPS with Dhh1-mCherry and can recapitulate *in vitro* the positive impact of Pat1 on LLPS formation, and provide evidence using different Pat1 variants that its interactions with Dhh1 and RNA are both important for stimulating LLPS formation. Finally, they determine that the stoichiometry of Pat1 relative to Dhh1 in the LLPS *in vitro* is similar to that in PBs *in vivo*.

This work is well performed with solid controls and clear results. It is an important study not only because it defines a direct role for an additional factor, Pat1, in the formation of PBs, and how it interplays with the other factors that regulate PB formation and disassembly, but also because it consolidates the ability of the *in vitro* system set up by the Weiss laboratory to study the dynamics of these bodies that play important roles in the regulation of gene expression. Nevertheless, there are a number of issues that require additional experimental data or revisions of text.

Essential revisions:

- It would useful to verify that the Pat1^4A-Dhh1^ and Pat1^EE^ variants are well expressed in cells to bolster the argument that these mutations primarily impair Pat1 function; although it is realized that failure to interact with Dhh1 or RNA might reduce the stability of these variants.

- It would be useful to extend the complementation analysis of the internally truncated PAT1-NC allele to a *pat1* mutant also lacking one or both of the EDC3 or SCD6 genes in order to provide a more rigorous test of its functionality *in vivo* compared to WT PAT1.

- It would be useful to demonstrate that a mutation in DHH1 that specifically impairs interaction with Pat1 (assuming one exists and has been rigorously shown to have this specific defect) also impairs PB formation to rule out the possibility that the Pat1^4A-Dhh1^ substitution impairs Pat1 binding to some other factor with a role in PB formation besides Dhh1.

- The model in Figure 6C is being questioned in that it lacks the LSm1-7 complex, whereas much of the available evidence indicates that Pat1 acts in the context of a Pat1/LSm complex to regulate mRNA turnover and translation in cells. In particular, there are data indicating that the presence of both Pat1 and Lsm proteins are required for efficient RNA association (RNA. 2014 Sep; 20(9): 1465-1475). To strengthen their model, it would be useful to show that a mutation in Pat1 that specifically impairs its RNA binding activity would reduce LLPS droplet assembly *in vitro*. While the Pat1^EE^ variant shows reduced RNA binding *in vitro* and also is defective for LLPS formation, it is unclear that this mutation specifically impairs LLPS formation by reducing RNA binding. It would strengthen their model if they could show that a mutation in the Pat1 C-domain that impairs RNA binding would also impair LLPS droplet formation. If such a Pat1 mutant has not been described in the literature then they could at least show that LLPS assembly is dependent on RNA and that single-stranded DNA will not suffice, to provide additional evidence that RNA is specifically required for LLPS assembly.

---

## [Author Response]

Essential revisions:- It would useful to verify that the Pat1^4A-Dhh1^ and Pat1^EE^ variants are well expressed in cells to bolster the argument that these mutations primarily impair Pat1 function; although it is realized that failure to interact with Dhh1 or RNA might reduce the stability of these variants.

As requested, we have measured the expression level of the different Pat1 mutants used in this study and compared it to the expression level of wild-type Pat1 via Western blotting (Figure 2—figure supplement 1). Both the Pat1^4A-Dhh1^ (Dhh1-binding mutant) and Pat1^EE^ (phospho-mimetic mutant) expressed as well as, if not better than wild-type Pat1 (Figure 2—figure supplement 1C). Moreover, we also checked the expression level of Dhh1 in the background of different Pat1 mutant cells compared to wild-type Pat1. Our data reveal that also Dhh1 is expressed to comparable levels in the different Pat1 backgrounds (Figure 2—figure supplement 1D).

- It would be useful to extend the complementation analysis of the internally truncated PAT1-NC allele to a pat1 mutant also lacking one or both of the EDC3 or SCD6 genes in order to provide a more rigorous test of its functionality *in vivo* compared to WT PAT1.

To further extend the analysis of the truncated Pat1 construct (Pat1-NC) and demonstrate its functionality *in vivo* we have assessed as requested its growth phenotype in different sensitized backgrounds under various temperatures. A *pat1* null mutant was shown to have a growth defect at a temperature of 37°C (Fourati et al., 2014) and Charenton et al., 2017 showed that a deletion of *PAT1* in an *edc3Δ scd6Δ* background also leads to a thermosensitive growth defect. We therefore employed these strains to test if the Pat1-NC construct is able to functionally replace wild-type Pat1. For this, we endogenously expressed Pat1-NC from its locus in *edc3Δ scd6Δ* cells and compared the growth phenotype to that of *edc3Δ scd6Δ* and *pat1Δ edc3Δ scd6Δ* cells at different temperatures. The Pat1-NC construct was able to rescue the growth phenotype of *pat1Δ edc3Δ scd6Δ* cells at the thermosensitive temperaturecomparable to wild-type, full-length Pat1 (Figure 1—figure supplement 1D).Overall, this demonstrates that the Pat1-NC construct is functional *in vivo*. In addition, we also show that the Pat1-NC construct rescues the growth defect of *pat1* null mutant cells at 37°C (Figure 1—figure supplement 1C).These new results together with the robust PB induction observed in Pat1-NC expressing cells (Figure 1C, D, E, F and Figure 1—figure supplement 1B, C) further bolster our conclusions that the Pat1-NC truncated construct behaves similar to wild-type Pat1 *in vivo*.

- It would be useful to demonstrate that a mutation in DHH1 that specifically impairs interaction with Pat1 (assuming one exists and has been rigorously shown to have this specific defect) also impairs PB formation to rule out the possibility that the Pat1^4A-Dhh1^ substitution impairs Pat1 binding to some other factor with a role in PB formation besides Dhh1.

We would like to thank the reviewers for this suggestion and have now added the analysis of PB formation in cells expressing mutants of Dhh1 that abolish binding to Pat1 (complementing our prior analysis with a Dhh1 binding mutant in Pat1 (Pat1^4A-Dhh1^). Sharif et al., 2013 previously defined -using structural data and binding studies- two patches on Dhh1, which when mutated block Pat1 binding. Dhh1-Mut-3A (Dhh1^S292DN294D^) weakens Dhh1-Pat1 binding whereas Dhh1-Mut-3B (Dhh1^R295D^) completely abolishes binding to Pat1.

Our data reveal that yeast cells expressing Dhh1^S292DN294D^ or Dhh1^R295D^ show defects in PB formation when compared to wild-type cells both in carbon starvation stress and upon hippuristanol treatment (Figure 3 and Videos 3, 4, 5). Remarkably, the Dhh1^R295D^ mutant that *in vitro* completely blocks Pat1 binding shows a more severe defect in PB formation in comparison to the partial Pat1 binder Dhh1^S292DN294D^(Figure 3). Both Dhh1^R295D^ and Dhh1^S292DN294D^ mutants expressed as well as wild-type Dhh1 (Figure 3—figure supplement 1). Taken together these new data strengthen our conclusion that a robust and direct interaction between Pat1 and Dhh1 is a key driver for PB formation in cells.

- The model in Figure 6C is being questioned in that it lacks the LSm1-7 complex, whereas much of the available evidence indicates that Pat1 acts in the context of a Pat1/LSm complex to regulate mRNA turnover and translation in cells. In particular, there are data indicating that the presence of both Pat1 and Lsm proteins are required for efficient RNA association (RNA. 2014 Sep; 20(9): 1465-1475). To strengthen their model, it would be useful to show that a mutation in Pat1 that specifically impairs its RNA binding activity would reduce LLPS droplet assembly *in vitro*. While the Pat1^EE^ variant shows reduced RNA binding *in vitro* and also is defective for LLPS formation, it is unclear that this mutation specifically impairs LLPS formation by reducing RNA binding. It would strengthen their model if they could show that a mutation in the Pat1 C-domain that impairs RNA binding would also impair LLPS droplet formation. If such a Pat1 mutant has not been described in the literature then they could at least show that LLPS assembly is dependent on RNA and that single-stranded DNA will not suffice, to provide additional evidence that RNA is specifically required for LLPS assembly.

As requested, we have now added new experiments to test the role of RNA in the *in vitro* liquid-liquid phase separation assay. Unfortunately, to our knowledge there is no Pat1 mutant described in the literature that has been shown to abolish its binding to RNA. We therefore tested the RNA-dependence of the Pat1-mediated enhancement of the phase separation of Dhh1. As shown before, in the absence of RNA (with ATP) no Dhh1 droplets are observed in our assay conditions [Mugler et al., 2016]. When increasing concentrations of Pat1 were added, we did not observe phase separation of either Dhh1 or Pat1 in the absence of RNA(Figure 6—figure supplement 2B) suggesting that the enhancement of phase separation of Dhh1 upon addition of Pat1 is strictly RNA-dependent. In order to further evaluate the critical role of RNA we also tested only the N-terminus of Pat1 lacking the C-terminal RNA-binding domain in the *in vitro* phase separation assay. Addition of increasing concentrations of Pat1-N to Dhh1 in the presence of RNA and ATP did not stimulate the phase separation of Dhh1 (Figure 6—figure supplement 2C) thus demonstrating that both the N and the C-terminus of Pat1 are essential to promote condensation and oligomerization of Dhh1 into liquid droplets. This suggests that the N-terminus of Pat1 binds Dhh1 directly and the C-terminus of Pat1 functions via RNA binding to promote higher order droplet formation.

We thank the reviewers for the suggestion to try the liquid-liquid phase separation assay also in the presence of ssDNA to further reveal the importance of RNA. However, we would like to point out that many DEAD-box ATPases are expected to bind to ssDNA via a phosphate-backbone interaction. For instance the DEAD-box ATPase Eukaryotic translation initiation factor 4A (eIF4A) was shown to bind ssDNA (Peck and Herschlag, 1999). Tai et al., 1996 showed that a DEAD-box protein of the hepatitis C virus can unwind DNA duplexes and in particular, we would like to draw the attention to the study by Dutta et al., 2011 where the authors demonstrate that Dhh1 is able to interact with ssDNA. We therefore did not include a phase separation assay with ssDNA, and we believe that the experiments described in Figure 6—figure supplement 2B, C serve as a better control.